# LAW OF BALANCE AND STATIONARY DISTRIBUTION OF STOCHASTIC GRADIENT DESCENT

## ABSTRACT

The stochastic gradient descent (SGD) algorithm is the algorithm we use to train neural networks. However, it remains poorly understood how the SGD navigates the highly nonlinear and degenerate loss landscape of a neural network. In this work, we prove that the minibatch noise of SGD regularizes the solution towards a balanced solution whenever the loss function contains a rescaling symmetry. Because the difference between a simple diffusion process and SGD dynamics is the most significant when symmetries are present, our theory implies that the loss function symmetries constitute an essential probe of how SGD works. We then apply this result to derive the stationary distribution of stochastic gradient flow for a diagonal linear network with arbitrary depth and width. The stationary distribution exhibits complicated nonlinear phenomena such as phase transitions, loss of ergodicity, and fluctuation inversion. These phenomena are shown to exist uniquely in deep networks, implying a fundamental difference between deep and shallow models.

## 1 INTRODUCTION

The stochastic gradient descent (SGD) algorithm is defined as

$$\Delta\theta_t = -\frac{\eta}{S}\sum_{x\in B}\nabla_\theta\ell(\theta, x), \tag{1}$$

where $\theta$ is the model parameter, $\ell(\theta, x)$ is a per-sample loss whose expectation over $x$ gives the training loss: $L(\theta) = \mathbb{E}_x[\ell(\theta, x)]$. $B$ is a randomly sampled minibatch of data points, each independently sampled from the training set, and $S$ is the minibatch size. Two aspects of the algorithm make it difficult to understand this algorithm: (1) its dynamics is discrete in time, and (2) the randomness is highly nonlinear and parameter-dependent. This work relies on the continuous-time approximation and deals with the second aspect.

In natural and social sciences, the most important object of study of a stochastic system is its stationary distribution, which is often found to offer fundamental insights into understanding a given stochastic process (Van Kampen, 1992; Rolski et al., 2009). Arguably, a great deal of insights into SGD can be obtained if we have an analytical understanding of its stationary distribution, which remains unknown until today. Existing works that study the dynamics and stationary properties of SGD are often restricted to the case of a strongly convex loss function (Wu et al., 2018; Xie et al., 2020; Liu et al., 2021; Zhu et al., 2018; Mori et al., 2022; Zou et al., 2021; Ma et al., 2018; Woodworth et al., 2020) or rely heavily on the local approximations of the stationary distribution of SGD close to a local minimum, often with additional unrealistic assumptions about the noise. For example, using a saddle point expansion and assuming that the noise is parameter-independent, a series of recent works showed that the stationary distribution of SGD is exponential Mandt et al. (2017); Xie et al. (2020); Liu et al. (2021). Taking partial parameter-dependence into account and near an interpolation minimum, Mori et al. (2022) showed that the stationary distribution is power-law-like and proportional to $L(\theta)^{-c_0}$ for some constant $c_0$. However, the stationary distribution of SGD is unknown when the loss function is beyond quadratic and high-dimensional.

Since the stationary distribution of SGD is unknown, we will compare our results with the most naive theory one can construct for SGD, a continuous-time Langevin equation with a constant noise level:

$$\dot{\theta}(t) = -\eta\nabla_\theta L(\theta) + \sqrt{2T_0}\epsilon(t), \tag{2}$$

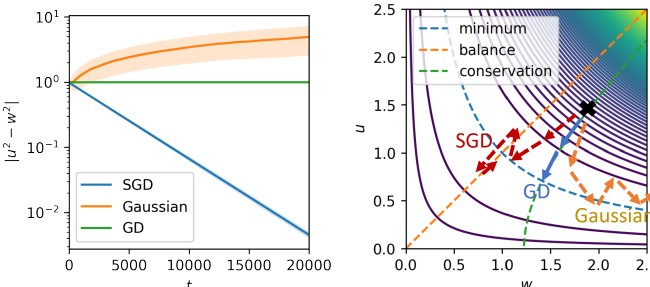

Figure 1: SGD converges to a balanced solution. **Left**: the quantity $u^2 - w^2$ is conserved for GD without noise, is divergent for GD with an isotropic Gaussian noise, which simulates the simple Langevin model, and decays to zero for SGD, making a sharp and dramatic contrast. **Right**: illustration of the three types of dynamics. Gradient descent (GD) moves along the conservation line due to the conservation law: $u^2(t) - w^2(t) = u^2(0) - w^2(0)$. GD with an isotropic Gaussian noise expands and diverges along the flat direction of the minimum valley. The actual SGD oscillates along a balanced solution.

where $\epsilon$ is a random time-dependent noise with zero mean and $\mathbb{E}[\epsilon(t)\epsilon(t')^T] = \eta\delta(t - t')I$ with $I$ being the identity operator. Here, the naive theory relies on the assumption that one can find a constant scalar $T_0$ such that Eq. (2) closely models (1), at least after some level of coarse-graining. Let us examine some of the predictions of this model to understand when and why it goes wrong.

There are two important predictions of this model. The first is that the stationary distribution of SGD is a Gibbs distribution with temperature $T_0$: $p(\theta) \propto \exp[-L(\theta)/T]$. This implies that the maximum likelihood estimator of $\theta$ under SGD is the same as the global minimizer of the $L(\theta)$: $\arg\max p(\theta) = \arg\min L(\theta)$. This relation holds for the local minima as well: every local minimum of $L$ corresponds to a local maximum of $p$. These properties are often required in the popular argument that SGD approximates Bayesian inference (Mandt et al., 2017; Mingard et al., 2021). Another implication is ergodicity (Walters, 2000): any state with the same energy will have an equal probability of being accessed. The second is the dynamical implication: SGD will *diffuse*. If there is a degenerate direction in the loss function, SGD will diffuse along that direction.[1]

However, these predictions of the Langevin model are not difficult to reject. Let us consider a simple two-layer network with the loss function: $\ell(u, w, x) = (uwx - y(x))^2$. Because of the rescaling symmetry, a valley of degenerate solution exists at $uw = const$. Under the simple Langevin model, SGD diverges to infinity due to diffusion. One can also see this from a static perspective. All points on the line $uw = c_0$ must have the same probability at stationarity, but such a distribution does not exist because it is not normalizable. This means that the Langevin model of SGD diverges for this loss function.

Does this agree with the empirical observation? Certainly not.[2] See Fig. 1. We see that contrary to the prediction of the Langevin model, $|u^2 - w^2|$ converges to zero under SGD. Under GD, this quantity is conserved during training (Du et al., 2018). Only the Gaussian GD obeys the prediction of the Langevin model, which is expected. This sharp contrast shows that the SGD dynamics is quite special, and a naive theoretical model can be very far from the truth in understanding its behavior. There is one more lesson to be learned. The fact that the Langevin model disagrees the most with the experiments when symmetry conditions are present suggests that the symmetry conditions are crucial tools to probe and understand the nature of the SGD noise, which is the main topic of our theory.

## 2  LAW OF BALANCE

Now, we consider the actual continuous-time limit of SGD (Latz, 2021; Li et al., 2019; 2021; Sirignano & Spiliopoulos, 2020; Fontaine et al., 2021; Hu et al., 2017):

$$d\theta = -\nabla_\theta L dt + \sqrt{TC(\theta)}dW_t, \tag{3}$$

where $dW_t$ is a stochastic process satisfying $dW_t \sim N(0, Idt)$ and $\mathbb{E}[dW_t dW_{t'}^T] = \delta(t - t')I$, and $T = \eta/S$. Apparently, $T$ gives the average noise level in the dynamics. Previous works have

---

[1]Note that this can also be seen as a dynamical interpretation of the ergodicity.

[2]In fact, had it been the case, no linear network or ReLU network can be trained with SGD.

suggested that the ratio $\eta/S \coloneqq T$ is the main factor determining the behavior of SGD, and a higher $T$ often leads to better generalization performance (Shirish Keskar et al., 2016; Liu et al., 2021; Ziyin et al., 2022b). The crucial difference between Eq. (3) and (2) is that in (3), the noise covariance $C(\theta)$ is parameter-dependent and, in general, low-rank when symmetries exist.

Due to standard architecture designs, a type of invariance – the rescaling symmetry – often appears in the loss function and exists for all sampling of minibatches. The per-sample loss $\ell$ is said to have the rescaling symmetry for all $x$ if $\ell(u, w, x) = \ell(\lambda u, w/\lambda, x)$ for a scalar $\lambda$ in an arbitrary neighborhood of 1.[3] This type of symmetry appears in many scenarios in deep learning. For example, it appears in any neural network with the ReLU activation. It also appears in the self-attention of transformers, often in the form of key and query matrices (Vaswani et al., 2017). When this symmetry exists between $u$ and $w$, one can prove the following result, which we refer to as the law of balance.

**Theorem 1.** (Law of balance.) *Let $u$ and $w$ be vectors of arbitrary dimensions. Let $\ell(u, w, x)$ satisfy $\ell(u, w, x) = \ell(\lambda u, w/\lambda, x)$ for arbitrary $x$ and any $\lambda$ in some neighborhood of 1. Then,*

$$\frac{d}{dt}(\|u\|^2 - \|w\|^2) = -T(u^T C_1 u - w^T C_2 w), \tag{4}$$

*where $C_1 = \mathbb{E}[A^T A] - \mathbb{E}[A^T]\mathbb{E}[A]$, $C_2 = \mathbb{E}[A A^T] - \mathbb{E}[A]\mathbb{E}[A^T]$ and $A_{ki} = \partial\tilde{\ell}/\partial(u_i w_k)$ with $\tilde{\ell}(u_i w_k, x) \equiv \ell(u_i, w_k, x)$.*

Our result holds in a stronger version if we consider the effect of a finite step-size by using the modified loss function (See Appendix B.7) (Barrett & Dherin, 2020; Smith et al., 2021). For common problems, $C_1$ and $C_2$ are positive definite, and this theorem implies that the norms of $u$ and $w$ will be approximately balanced. To see this, one can identify its upper and lower bounds:

$$-T(\lambda_{1M}\|u\|^2 - \lambda_{2m}\|w\|^2) \leq \frac{d}{dt}(\|u\|^2 - \|w\|^2) \leq -T(\lambda_{1m}\|u\|^2 - \lambda_{2M}\|w\|^2), \tag{5}$$

where $\lambda_{1m(2m)}, \lambda_{1M(2M)}$ represent the minimal and maximal eigenvalue of the matrix $C_{1(2)}$, respectively. In the long-time limit, the value of $\|u\|^2/\|w\|^2$ is restricted by

$$\frac{\lambda_{2m}}{\lambda_{1M}} \leq \frac{\|u\|^2}{\|w\|^2} \leq \frac{\lambda_{2M}}{\lambda_{1m}}, \tag{6}$$

which implies that the stationary dynamics of the parameters $u, w$ is constrained in a bounded subspace of the unbounded degenerate local minimum valley. Conventional analysis shows that the difference between SGD and GD is of order $T^2$ per unit time step, and it is thus often believed that SGD can be understood perturbatively through GD (Hu et al., 2017). However, the law of balance implies that the difference between GD and SGD is not perturbative. As long as there is any level of noise, the difference between GD and SGD at stationarity is $O(1)$. This theorem also implies the loss of ergodicity, an important phenomenon in nonequilibrium physics (Palmer, 1982; Thirumalai & Mountain, 1993; Mauro et al., 2007; Turner et al., 2018), because not all solutions with the same training loss will be accessed by SGD with equal probability.[4]

The theorem greatly simplifies when both $u$ and $w$ are one-dimensional.

**Corollary 1.** *If $u, w \in \mathbb{R}$, then, $\frac{d}{dt}|u^2 - w^2| = -TC_0|u^2 - w^2|$, where $C_0 = \text{Var}[\frac{\partial\ell}{\partial(uw)}]$.*

Before we apply the theorem to study the stationary distributions, we stress the importance of this balance condition. This relation is closely related to Noether's theorem (Misawa, 1988; Baez & Fong, 2013; Malinowska & Ammi, 2014). If there is no weight decay or stochasticity in training, the quantity $\|u\|^2 - \|w\|^2$ will be a conserved quantity under gradient flow (Du et al., 2018; Kunin et al., 2020), as is evident by taking the infinite $S$ limit. The fact that it monotonically decays to zero at a finite $T$ may be a manifestation of some underlying fundamental mechanism. A more recent result by Wang et al. (2022) showed that for a two-layer linear network, the norms of two layers are within a distance of order $O(\eta^{-1})$, suggesting that the norm of the two layers are balanced. Our

---

[3]Note that this is a weaker condition than the common definition of rescaling symmetry, where the condition holds for an arbitrary positive $\lambda$.

[4]This could imply that SGD has a high efficacy at exploring a high-dimensional landscape because the degenerate symmetry directions are essentially ignored during the exploration.

result agrees with Wang et al. (2022) in this case, but our result is stronger because our result is nonperturbative, only relies on the rescaling symmetry, and is independent of the loss function or architecture of the model. It is useful to note that when $L_2$ regularization with strength $\gamma$ is present, the rate of decay changes from $TC_0$ to $TC_0 + \gamma$. This gives us a nice interpretation that when rescaling symmetry is present, the implicit bias of SGD is equivalent to weight decay.

**Example: two-layer linear network.** It is instructive to illustrate the application of the law to a two-layer linear network, the simplest model that obeys the law. Let $\theta = (w, u)$ denote the set of trainable parameters; the per-sample loss is $\ell(\theta, x) = (\sum_i^d u_i w_i x - y)^2 + \gamma\|\theta\|^2$. Here, $d$ is the width of the model, $\gamma\|\theta\|^2$ is the $L_2$ regularization term with strength $\gamma \geq 0$, and $\mathbb{E}_x$ denotes the averaging over the training set, which could be a continuous distribution or a discrete sum of delta distributions. It will be convenient for us also to define the shorthand: $v := \sum_i^d u_i w_i$. The distribution of $v$ is said to be the distribution of the "model."

Applying the law of balance, we obtain that

$$\frac{d}{dt}(u_i^2 - w_i^2) = -4[T(\alpha_1 v^2 - 2\alpha_2 v + \alpha_3) + \gamma](u_i^2 - w_i^2), \tag{7}$$

where we have introduced the parameters

$$\begin{cases} \alpha_1 := \text{Var}[x^2], \\ \alpha_2 := \mathbb{E}[x^3 y] - \mathbb{E}[x^2]\mathbb{E}[xy], \\ \alpha_3 := \text{Var}[xy]. \end{cases} \tag{8}$$

When $\alpha_1\alpha_3 - \alpha_2^2$ or $\gamma > 0$, the time evolution of $|u^2 - w^2|$ can be upper-bounded by an exponentially decreasing function in time: $|u_i^2 - w_i^2|(t) < |u_i^2 - w_i^2|(0) \exp\left(-4T(\alpha_1\alpha_3 - \alpha_2^2)t/\alpha_1 - 4\gamma t\right) \to 0$. Namely, the quantity $(u_i^2 - w_i^2)$ decays to $0$ with probability $1$. We thus have $u_i^2 = w_i^2$ for all $i \in \{1, \cdots, d\}$ at stationarity, in agreement with what we see in Figure 1.

## 3 STATIONARY DISTRIBUTION OF SGD

As an important application of the law of balance, we solve the stationary distribution of SGD for a deep diagonal linear network. While linear networks are limited in expressivity, their loss landscape and dynamics are highly nonlinear and is regarded as a minimal model of nonlinear neural networks (Kawaguchi, 2016; Saxe et al., 2013; Ziyin et al., 2022a).

### 3.1 DEPTH-0 CASE

Let us first derive the stationary distribution of a one-dimensional linear regressor, which will be a basis for comparison to help us understand what is unique about having a "depth" in deep learning. The per-sample loss is $\ell(x, v) = (vx - y)^2 + \gamma v^2$, for which the SGD dynamics is $dv = -2(\beta_1 v - \beta_2 + \gamma v)dt + \sqrt{TC(v)}dW(t)$, where we have defined

$$\begin{cases} \beta_1 := \mathbb{E}[x^2], \\ \beta_2 := \mathbb{E}[xy]. \end{cases} \tag{9}$$

Note that the closed-form solution of linear regression gives the global minimizer of the loss function: $v^* = \beta_2/\beta_1$. The gradient variance is also not trivial: $C(v) := \text{Var}[\nabla_v \ell(v, x)] = 4(\alpha_1 v^2 - 2\alpha_2 v + \alpha_3)$. Note that the loss landscape $L$ only depends on $\beta_1$ and $\beta_2$, and the gradient noise only depends on $\alpha_1$, $\alpha_2$ and, $\alpha_3$. These relations imply that $C$ can be quite independent of $L$, contrary to popular beliefs in the literature (Mori et al., 2022; Mandt et al., 2017). Here independence between $C$ and $L$ comes from the fact that the noise only depends on the variance of the data $x^2$ and $xy$ while $L$ only depends on the expectation of the data. It is also reasonable to call $\beta$ the landscape parameters and $\alpha$ the noise parameters. We will see that both $\beta$ and $\alpha$ are important parameters appearing in all stationary distributions we derive, implying that the stationary distributions of SGD are strongly dependent on the data.

Another important quantity is $\Delta := \min_v C(v) \geq 0$, which is the minimal level of noise on the landscape. For all the examples in this work,

$$\Delta = \text{Var}[x^2]\text{Var}[xy] - \text{cov}(x^2, xy) = \alpha_1\alpha_3 - \alpha_2^2. \tag{10}$$

When is $\Delta$ zero? It happens when, for all samples of $(x, y)$, $xy + c = kx^2$ for some constant $k$ and $c$. We focus on the case $\Delta > 0$ in the main text, which is most likely the case for practical situations. The other cases are dealt with in Section B.

For $\Delta > 0$, the stationary distribution for linear regression is found to be

$$p(v) \propto (\alpha_1 v^2 - 2\alpha_2 v + \alpha_3)^{-1-\frac{\beta_1'}{2T\alpha_1}} \exp\left[-\frac{1}{T}\frac{\alpha_2\beta_1' - \alpha_1\beta_2}{\alpha_1\sqrt{\Delta}}\arctan\left(\frac{\alpha_1 v - \alpha_2}{\sqrt{\Delta}}\right)\right], \quad (11)$$

roughly in agreement with the result in Mori et al. (2022). Two notable features exist for this distribution: (1) the power exponent for the tail of the distribution depends on the learning rate and batch size, and (2) the integral of $p(v)$ converges for an arbitrary learning rate. On the one hand, this implies that increasing the learning rate alone cannot introduce new phases of learning to a linear regression; on the other hand, it implies that the expected error is divergent as one increases the learning rate (or the feature variation), which happens at $T = \beta_1'/\alpha_1$. We will see that deeper models differ from the single-layer model in these two crucial aspects.

## 3.2 Deep Diagonal Networks

Now, we consider a diagonal deep linear network, whose loss function can be written as

$$\ell = \left[\sum_i^{d_0}\left(\prod_{k=0}^{D} u_i^{(k)}\right)x - y\right]^2, \quad (12)$$

where $D$ is the depth and $d_0$ is the width. When the width $d_0 = 1$, the law of balance is sufficient to solve the model. When $d_0 > 1$, we need to eliminate additional degrees of freedom. A lot of recent works study the properties of a diagonal linear network, which has been found to well approximate the dynamics of real networks (Pesme et al., 2021; Nacson et al., 2022; Berthier, 2023; Even et al., 2023).

We introduce $v_i := \prod_{k=0}^{D} u_i^{(k)}$, and so $v = \sum_i v_i$, where we call $v_i$ a "subnetwork" and $v$ the "model." The following proposition shows that the dynamics of this model can be reduced to a one-dimensional form.

**Theorem 2.** *For all $i \neq j$, one (or more) of the following conditions holds for all trajectories at stationarity:*

1. *$v_i = 0$, or $v_j = 0$, or $L(\theta) = 0$;*
2. *$\text{sgn}(v_i) = \text{sgn}(v_j)$. In addition, (a) if $D = 1$, for a constant $c_0$, $\log|v_i| - \log|v_j| = c_0$; (b) if $D > 1$, $|v_i|^2 - |v_j|^2 = 0$.*

This theorem contains many interesting aspects. First of all, the three situations in item 1 directly tell us the distribution of $v$, which is the quantity we ultimately care about.[5] This result implies that if we want to understand the stationary distribution of SGD, we only need to solve the case of item 2. Once the parameters enter the condition of item 2, item 2 will continue to hold with probability 1 for the rest of the trajectory. The second aspect is that item 2 of the theorem implies that all the $v_i$ of the model must be of the same sign for any network with $D \geq 1$. Namely, no subnetwork of the original network can learn an incorrect sign. This is dramatically different from the case of $D = 0$. We will discuss this point in more detail below. The third interesting aspect of the theorem is that it implies that the dynamics of SGD is qualitatively different for different depths of the model. In particular, $D = 1$ and $D > 1$ have entirely different dynamics. For $D = 1$, the ratio between every pair of $v_i$ and $v_j$ is a conserved quantity. In sharp contrast, for $D > 1$, the distance between different $v_i$ is no longer conserved but decays to zero. Therefore, a new balancing condition emerges as we increase the depth. Conceptually, this qualitative distinction also corroborates the discovery in Ziyin et al. (2022a) and Ziyin & Ueda (2022), where $D = 1$ models are found to be qualitatively different from models with $D > 1$.

With this theorem, we are now ready to solve for the stationary distribution. It suffices to condition on the event that $v_i$ does not converge to zero. Let us suppose that there are $d$ nonzero $v_i$ that obey item 2 of Theorem 2 and $d$ can be seen as an effective width of the model. We stress that the effective

---

[5]$L \to 0$ is only possible when $\Delta = 0$ *and* $v = \beta_2/\beta_1$.

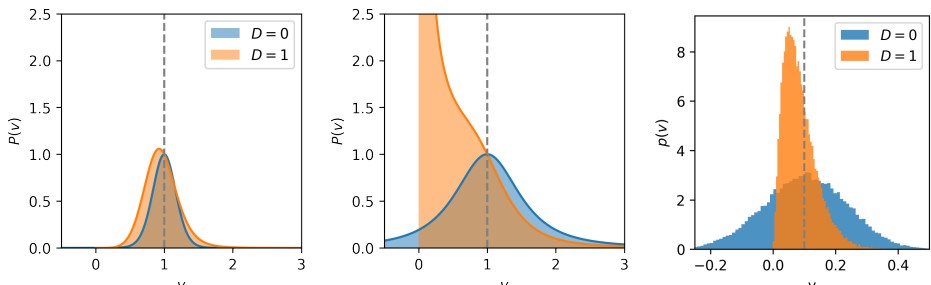

Figure 2: Stationary distributions of SGD for simple linear regression ($D = 0$), and a two-layer network ($D = 1$) across different $T = \eta/S$: $T = 0.05$ (**left**) and $T = 0.5$ (**Mid**). We see that for $D = 1$, the stationary distribution is strongly affected by the choice of the learning rate. In contrast, for $D = 0$, the stationary distribution is also centered at the global minimizer of the loss function, and the choice of the learning rate only affects the thickness of the tail. **Right**: the stationary distribution of a one-layer $\tanh$-model, $f(x) = \tanh(vx)$ ($D = 0$) and a two-layer $\tanh$-model $f(x) = w\tanh(ux)$ ($D = 1$). For $D = 1$, we define $v := wu$. The vertical line shows the ground truth. The deeper model never learns the wrong sign of $wu$, whereas the shallow model can learn the wrong one.

width $d \le d_0$ depends on the initialization and can be arbitrary.[6] Therefore, we condition on a fixed value of $d$ to solve for the stationary distribution of $v$ (Appendix B):

$$p_{\pm}(|v|) \propto \frac{1}{|v|^{3(1-1/(D+1))}(\alpha_1|v|^2 \mp 2\alpha_2|v| + \alpha_3)} \exp\left(-\frac{1}{T}\int_0^{|v|} d|v| \frac{d^{1-2/(D+1)}(\beta_1|v| \mp \beta_2)}{(D+1)|v|^{2D/(D+1)}(\alpha_1|v|^2 \mp 2\alpha_2|v| + \alpha_3)}\right), \tag{13}$$

where $p_-$ is the distribution of $v$ on $(-\infty, 0)$ and $p_+$ is that on $(0, \infty)$. Next, we analyze this distribution in detail. Since the result is symmetric in the sign of $\beta_2 = \mathbb{E}[xy]$, we assume that $\mathbb{E}[xy] > 0$ from now on.

### 3.2.1 Depth-1 Nets

We focus on the case $\gamma = 0$.[7] The distribution of $v$ is

$$p_{\pm}(|v|) \propto \frac{|v|^{\pm\beta_2/2\alpha_3 T - 3/2}}{(\alpha_1|v|^2 \mp 2\alpha_2|v| + \alpha_3)^{1 \pm \beta_2/4T\alpha_3}} \exp\left(-\frac{1}{2T}\frac{\alpha_3\beta_1 - \alpha_2\beta_2}{\alpha_3\sqrt{\Delta}} \arctan\frac{\alpha_1|v| \mp \alpha_2}{\sqrt{\Delta}}\right), \tag{14}$$

This measure is worth a close examination. First, the exponential term is upper and lower bounded and well-behaved in all situations. In contrast, the polynomial term becomes dominant both at infinity and close to zero. When $v < 0$, the distribution is a delta function at zero: $p(v) = \delta(v)$. To see this, note that the term $v^{-\beta_2/2\alpha_3 T - 3/2}$ integrates to give $v^{-\beta_2/2\alpha_3 T - 1/2}$ close to the origin, which is infinite. Away from the origin, the integral is finite. This signals that the only possible stationary distribution has a zero measure for $v \ne 0$. The stationary distribution is thus a delta distribution, meaning that if $x$ and $y$ are positively correlated, the learned subnets $v_i$ can never be negative, no matter the initial configuration.

For $v > 0$, the distribution is nontrivial. Close to $v = 0$, the distribution is dominated by $v^{\beta_2/2\alpha_3 T - 3/2}$, which integrates to $v^{\beta_2/2\alpha_3 T - 1/2}$. It is only finite below a critical $T_c = \beta_2/\alpha_3$. This is a phase-transition-like behavior. As $T \to (\beta_2/\alpha_3)_-$, the integral diverges and tends to a delta distribution. Namely, if $T > T_c$, we have $u_i = w_i = 0$ for all $i$ with probability 1, and no learning can happen. If $T < T_c$, the stationary distribution has a finite variance, and learning may happen. In the more general setting, where weight decay is present, this critical $T$ shifts to

$$T_c = \frac{\beta_2 - \gamma}{\alpha_3}. \tag{15}$$

When $T = 0$, the phase transition occurs at $\beta_2 = \gamma$, in agreement with the threshold weight decay identified in Ziyin & Ueda (2022). This critical learning rate also agrees with the discrete-time

---

[6]One can systematically initialize the parameters in a way that $d$ takes any desired value between 1 and $d_0$; for example, one way to achieve this is to initialize on the stationary conditions specified by Theorem 2 at the desired value of $d$.

[7]When weight decay is present, the stationary distribution is the same, except that one needs to replace $\beta_2$ with $\beta_2 - \gamma$. Other cases are also studied in detail in Appendix B and listed in Table. 1.

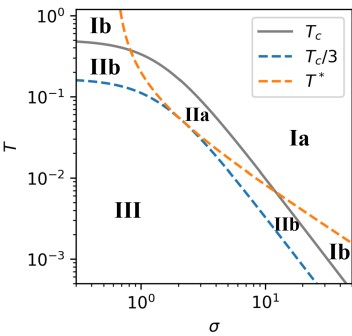

Figure 3: Regimes of learning for SGD as a function of $T = \eta/S$ and the noise in the dataset $\sigma$ for the noisy Gaussian dataset. According to (1) whether the sparse transition has happened, (2) whether a nontrivial maximum probability estimator exists, and (3) whether the sparse solution is a maximum probability estimator, the learning of SGD can be characterized into 5 regimes. Regime **I** is where SGD converges to a sparse solution with zero variance. In regime **II**, the stationary distribution has a finite spread, and the probability density of the sparse solution diverges. Hence, the probability of being close to the sparse solution is very high. In regime **III**, the probability density of the sparse solution is zero, and therefore the model will learn without much problem. In regime **b**, a local nontrivial probability maximum exists, and hence SGD has some probability of successful learning. The only maximum probability estimator in regime **a** is the sparse solution.

analysis performed in Ziyin et al. (2021; 2023) and the approximate continuous-time analysis in Chen et al. (2023). See Figure 2 for illustrations of the distribution across different values of $T$. We also compare with the stationary distribution of a depth-0 model. Two characteristics of the two-layer model appear rather striking: (1) the solution becomes a delta distribution at the sparse solution $u = w = 0$ at a large learning rate; (2) the two-layer model never learns the incorrect sign ($v$ is always non-negative). See Figure 2.

Therefore, training with SGD on deeper models simultaneously has two advantages: (1) a generalization advantage such that a sparse solution is favored when the underlying data correlation is weak; (2) an optimization advantage such that the training loss interpolates between that of the global minimizer and the sparse saddle and is well-bounded (whereas a depth-0 model can have arbitrarily bad objective value at a large learning rate).

Another exotic phenomenon implied by the result is what we call the "fluctuation inversion." Naively, the variance of model parameters should increase as we increase $T$, which is the noise level in SGD. However, for the distribution we derived, the variance of $v$ and $u$ both decrease to zero as we increase $T$: injecting noise makes the model fluctuation vanish. We discuss more about this "fluctuation inversion" in the next section.

Also, while there is no other phase-transition behavior below $T_c$, there is still an interesting and practically relevant crossover behavior in the distribution of the parameters as we change the learning rate. When we train a model, we often run SGD only once or a few times. When we do this, the most likely parameter we obtain is given by the maximum likelihood estimator of the distribution, $\hat{v} := \arg\max p(v)$. Understanding how $\hat{v}(T)$ changes as a function of $T$ is crucial. This quantity also exhibits nontrivial crossover behaviors at critical values of $T$.

When $T < T_c$, a nonzero maximizer for $p(v)$ must satisfy

$$v^* = -\frac{\beta_1 - 10\alpha_2 T - \sqrt{(\beta_1 - 10\alpha_2 T)^2 + 28\alpha_1 T(\beta_2 - 3\alpha_3 T)}}{14\alpha_1 T}. \tag{16}$$

The existence of this solution is nontrivial, which we analyze in Appendix B.5. When $T \to 0$, a solution always exists and is given by $v = \beta_2/\beta_1$, which does not depend on the learning rate or noise $C$. Note that $\beta_2/\beta_1$ is also the minimum point of $L(u_i, w_i)$. This means that SGD is only a consistent estimator of the local minima in deep learning in the vanishing learning rate limit. How biased is SGD at a finite learning rate? Two limits can be computed. For a small learning rate, the leading order correction to the solution is $v = \frac{\beta_2}{\beta_1} + \left(\frac{10\alpha_2\beta_2}{\beta_1^2} - \frac{7\alpha_1\beta_2^2}{\beta_1^3} - \frac{3\alpha_3}{\beta_1}\right)T$. This implies that the common Bayesian analysis that relies on a Laplace expansion of the loss fluctuation around a local minimum is improper. The fact that the stationary distribution of SGD is very far away from the Bayesian posterior also implies that SGD is only a good Bayesian sampler at a small learning rate.

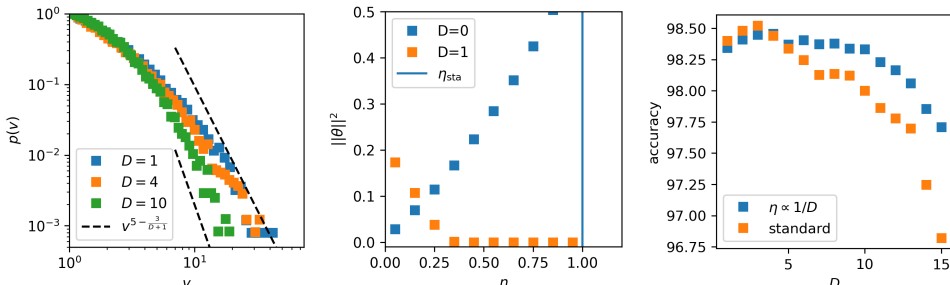

Figure 4: SGD on deep networks leads to a well-controlled distribution and training loss. **Left**: Power law of the tail of the parameter distribution of deep linear nets. The dashed lines show the upper $(-7/2)$ and lower $(-5)$ bound of the exponent of the tail. The predicted power-law scaling agrees with the experiment, and the exponent decreases as the theory predicts. **Mid**: training loss of a tanh network. $D = 0$ is the case where only the input weight is trained, and $D = 1$ is the case where both input and output layers are trained. For $D = 0$, the model norm increases as the model loses stability. For $D = 1$, a "fluctuation inversion" effect appears. The fluctuation of the model vanishes before it loses stability. **Right**: performance of fully connected tanh nets on MNIST. Scaling the learning rate as $1/D$ keeps the model performance relatively unchanged.

It is instructive to consider an example of a structured dataset: $y = kx + \epsilon$, where $x \sim \mathcal{N}(0, 1)$ and the noise $\epsilon$ obeys $\epsilon \sim \mathcal{N}(0, \sigma^2)$. We let $\gamma = 0$ for simplicity. If $\sigma^2 > \frac{8}{21}k^2$, there always exists a transitional learning rate: $T^* = \frac{4k + \sqrt{42}\sigma}{4(21\sigma^2 - 8k^2)}$. Obviously, $T_c/3 < T^*$. One can characterize the learning of SGD by comparing $T$ with $T_c$ and $T^*$. For this simple example, SGD can be classified into roughly 5 different regimes. See Figure 3.

### 3.3 POWER-LAW TAIL OF DEEPER MODELS

An interesting aspect of the depth-1 model is that its distribution is independent of the width $d$ of the model. This is not true for a deep model, as seen from Eq. (13). The $d$-dependent term vanishes only if $D = 1$. Another intriguing aspect of the depth-1 distribution is that its tail is independent of any hyperparameter of the problem, dramatically different from the linear regression case. This is true for deeper models as well.

Since $d$ only affects the non-polynomial part of the distribution, the stationary distribution scales as $p(v) \propto \frac{1}{v^{3(1-1/(D+1))}(\alpha_1 v^2 - 2\alpha_2 v + \alpha_3)}$. Hence, when $v \to \infty$, the scaling behaviour is $v^{-5+3/(D+1)}$. The tail gets monotonically thinner as one increases the depth. For $D = 1$, the exponent is $7/2$; an infinite-depth network has an exponent of $5$. Therefore, the tail of the model distribution only depends on the depth and is independent of the data or details of training, unlike the depth-0 model. In addition, due to the scaling $v^{5-3/(D+1)}$ for $v \to \infty$, we can see that $\mathbb{E}[v^2]$ will never diverge no matter how large the $T$ is. See Figure 4–mid.

One implication is that neural networks with at least one hidden layer will never have a divergent training loss. This directly explains the puzzling observation of the edge-of-stability phenomenon in deep learning: SGD training often gives a neural network a solution where a slight increment of the learning rate will cause discrete-time instability and divergence Wu et al. (2018); Cohen et al. (2021). These solutions, quite surprisingly, exhibit low training and testing loss values even when the learning rate is right at the critical learning rate of instability. This observation contradicts naive theoretical expectations. Let $\eta_{\text{sta}}$ denote the largest stable learning rate. Close to a local minimum, one can expand the loss function up to the second order to show that the value of the loss function $L$ is proportional to $\text{Tr}[\Sigma]$. However, $\Sigma \propto 1/(\eta_{\text{sta}} - \eta)$ should be a very large value (Yaida, 2018; Ziyin et al., 2022b; Liu et al., 2021), and therefore $L$ should diverge. Thus, the edge of stability phenomenon is incompatible with the naive expectation up to the second order, as pointed out in Damian et al. (2022). Our theory offers a direct explanation of why the divergence of loss does not happen: for deeper models, the fluctuation of model parameters decreases as the gradient noise level increases, reaching a minimal value before losing stability. Thus, SGD always has a finite loss because of the power-law tail and fluctuation inversion.

### 3.4 ROLE OF WIDTH

As discussed, for $D > 1$, the model width $d$ directly affects the stationary distribution of SGD. However, the integral in the exponent of Eq. (13) cannot be analytically calculated for a generic $D$.

Two cases exist where an analytical solution exists: $D = 1$ and $D \to \infty$. We thus consider the case $D \to \infty$ to study the effect of $d$.

As $D$ tends to infinity, the distribution becomes

$$p(v) \propto \frac{1}{v^{3+k_1}(\alpha_1 v^2 - 2\alpha_2 v + \alpha_3)^{1-k_1/2}} \exp\left(-\frac{d}{DT}\left(\frac{\beta_2}{\alpha_3 v} + \frac{\alpha_2\alpha_3\beta_1 - 2\alpha_2^2\beta_2 + \alpha_1\alpha_3\beta_2}{\alpha_3^2\sqrt{\Delta}}\arctan(\frac{\alpha_1 v - \alpha_2}{\sqrt{\Delta}}))\right)\right),$$

(17)

where $k_1 = d(\alpha_3\beta_1 - 2\alpha_2\beta_2)/(TD\alpha_3^2)$. The first striking feature is that the architecture ratio $d/D$ always appears simultaneously with $1/T$. This implies that for a sufficiently deep neural network, the ratio $D/d$ also becomes proportional to the strength of the noise. Since we know that $T = \eta/S$ determines the performance of SGD, our result thus shows an extended scaling law of training: $\frac{d}{D}\frac{S}{\eta} = const$. For example, if we want to scale up the depth without changing the width, we can increase the learning rate proportionally or decrease the batch size. This scaling law thus links all the learning rates, the batch size, and the model width and depth. The architecture aspect of the scaling law also agrees with an alternative analysis (Hanin, 2018; Hanin & Rolnick, 2018), where the optimal architecture is found to have a constant ratio of $d/D$. See Figure 4.

Now, we fix $T$ and understand the infinite depth limits, which is decided by the scaling of $d$ as we scale up $D$. There are three situations: (1) $d = o(D)$, (2) $d = c_0 D$ for a constant $c_0$, (3) $d = \Omega(D)$. If $d = o(D)$, $k_1 \to 0$ and the distribution converges to $p(v) \propto v^{-3}(\alpha_1 v^2 - 2\alpha_2 v + \alpha_3)^{-1}$, which is a delta distribution at $0$. Namely, if the width is far smaller than the depth, the model will collapse, and no learning will happen under SGD. Therefore, we should increase the model width as we increase the depth. In the second case, $d/D$ is a constant and can thus be absorbed into the definition of $T$ and is the only limit where we obtain a nontrivial distribution with a finite spread. If $d = \Omega(D)$, one can perform a saddle point approximation to see that the distribution becomes a delta distribution at the global minimum of the loss landscape, $p(v) = \delta(v - \beta_2/\beta_1)$. Therefore, the learned model locates deterministically at the global minimum.

## 4 DISCUSSION

The first implication of our theory is that the behavior of SGD cannot be understood through gradient flow or a simple Langevin approximation. Having a perturbative amount of noise in SGD leads to an order-1 change in the stationary solution. This suggests that one promising way to understand SGD is to study its behavior on a landscape from the viewpoint of symmetries. We showed that SGD systematically moves towards a balanced solution when rescaling symmetry exists. Likewise, it is not difficult to imagine that for other symmetries, SGD will also have order-1 deviations from gradient flow. An important future direction is thus to characterize the SGD dynamics on a loss function with other symmetries.

Using the symmetry conditions, we have characterized the stationary distribution of SGD analytically. To the best of our knowledge, this is the first analytical expression for a globally

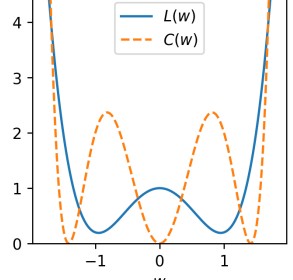

Figure 5: Loss landscape and noise covariance of a one-neuron linear network with one hidden neuron and $\gamma = 0.005$. The orange curve shows the noise covariance $C(w, u)$ for $w = u$. The shape of the gradient noise is, in general, more complicated than the landscape.

nonconvex and highly nonlinear loss without the need for any approximation. With this solution, we have discovered many phenomena of deep learning that were previously unknown. For example, we showed the qualitative difference between networks with different depths, the fluctuation inversion effect, the loss of ergodicity, and the incapability of learning a wrong sign for a deep model.

Lastly, let us return to the starting question: when is the Gibbs measure a bad model of SGD? When the number of data points $N \gg S$, a standard computation shows that the noise covariance of SGD takes the following form:$C(\theta) = T(\mathbb{E}_x[(\nabla_\theta\ell)(\nabla_\theta\ell)^T] - (\nabla_\theta L)(\nabla_\theta L)^T)$, which is nothing but the covariance of the gradients of $\theta$. A key feature of the noise is that it depends on the dynamical variable $\theta$ in a highly nontrivial manner (For example, see Hodgkinson & Mahoney (2021)). Alternatively, one can also understand this problem from the modified loss perspective (Geiping et al., 2021)). See Figure 5 for an illustration of the landscape against $C$. We see that the shape of $C(\theta)$ generally changes faster than the loss landscape. For the Gibbs distribution to hold (at least locally), we need $C(\theta)$ to change much slower than $L(\theta)$.

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

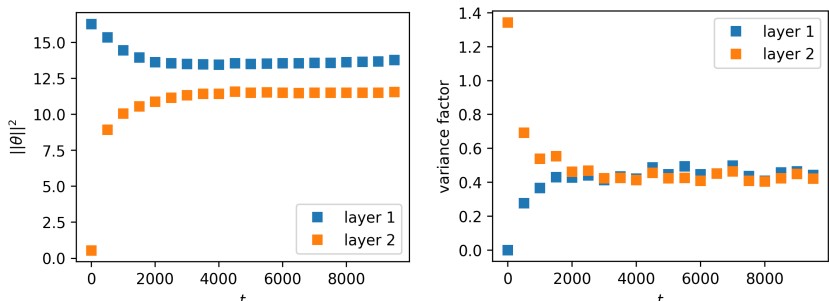

Figure 6: A two-layer ReLU network trained on full-rank dataset. **Left**: because of the rescaling symmetry, the norm of the two layers are balanced approximately. **Right**: the first and second terms in Eq. 4. We see that both terms converge to a stationary time-independent constant.

# A  ADDITIONAL EXPERIMENTS

## A.1  LAW OF BALANCE

See Figure 6. We train a two-layer ReLU network with the number of neurons: $20 \rightarrow 200 \rightarrow 20$. The dataset is a synthetic data set, where $x$ is drawn from a normal distribution, and the labels: $y = x + \epsilon$, for an independent Gaussian noise $\epsilon$ with unit variance.

## B  THEORETICAL CONSIDERATIONS

### B.1  BACKGROUND

#### B.1.1  ITO'S LEMMA

Let us consider the following stochastic differential equation (SDE) for a Wiener process $W(t)$:

$$dX_t = \mu_t dt + \sigma_t dW(t). \tag{18}$$

We are interested in the dynamics of a generic function of $X_t$. Let $Y_t = f(t, X_t)$; Ito's lemma states that the SDE for the new variable is

$$df(t, X_t) = \left( \frac{\partial f}{\partial t} + \mu_t \frac{\partial f}{\partial X_t} + \frac{\sigma_t^2}{2} \frac{\partial^2 f}{\partial X_t^2} \right) dt + \sigma_t \frac{\partial f}{\partial x} dW(t). \tag{19}$$

Let us take the variable $Y_t = X_t^2$ as an example. Then the SDE is

$$dY_t = \left( 2\mu_t X_t + \sigma_t^2 \right) dt + 2\sigma_t X_t dW(t). \tag{20}$$

Let us consider another example. Let two variables $X_t$ and $Y_t$ follow

$$\begin{aligned} dX_t &= \mu_t dt + \sigma_t dW(t), \\ dY_t &= \lambda_t dt + \phi_t dW(t). \end{aligned} \tag{21}$$

The SDE of $X_t Y_t$ is given by

$$d(X_t Y_t) = (\mu_t Y_t + \lambda_t X_t + \sigma_t \phi_t) dt + (\sigma_t Y_t + \phi_t X_t) dW(t). \tag{22}$$

#### B.1.2  FOKKER PLANCK EQUATION

The general SDE of a 1d variable $X$ is given by:

$$dX = -\mu(X) dt + B(X) dW(t). \tag{23}$$

The time evolution of the probability density $P(x,t)$ is given by the Fokker-Planck equation:

$$\frac{\partial P(X,t)}{\partial t} = -\frac{\partial}{\partial X} J(X,t), \tag{24}$$

where $J(X,t) = \mu(X)P(X,t) + \frac{1}{2}\frac{\partial}{\partial X}[B^2(X)P(X,t)]$. The stationary distribution satisfying $\partial P(X,t)/\partial t = 0$ is

$$P(X) \propto \frac{1}{B^2(X)} \exp\left[ -\int dX \frac{2\mu(X)}{B^2(X)} \right] := \tilde{P}(X), \tag{25}$$

which gives a solution as a Boltzmann-type distribution if $B$ is a constant. We will apply Eq. (25) to determine the stationary distributions in the following sections.

### B.2  PROOF OF THEOREM 1

*Proof.* By definition of the symmetry $\ell(\mathbf{u}, \mathbf{w}, x) = \ell(\lambda \mathbf{u}, \mathbf{w}/\lambda, x)$, we obtain its infinitesimal transformation $\ell(\mathbf{u}, \mathbf{w}, x) = \ell((1 + \epsilon)\mathbf{u}, (1 - \epsilon)\mathbf{w}/\lambda, x)$. Expanding this to first order in $\epsilon$, we obtain

$$\sum_i u_i \frac{\partial \ell}{\partial u_i} = \sum_j w_j \frac{\partial \ell}{\partial w_j}. \tag{26}$$

The equations of motion are

$$\frac{du_i}{dt} = -\frac{\partial \ell}{\partial u_i}, \tag{27}$$

$$\frac{dw_j}{dt} = -\frac{\partial \ell}{\partial w_j}. \tag{28}$$

Using Ito's lemma, we can find the equations governing the evolutions of $u_i^2$ and $w_j^2$:

$$\frac{du_i^2}{dt} = 2u_i \frac{du_i}{dt} + \frac{(du_i)^2}{dt} = -2u_i \frac{\partial \ell}{\partial u_i} + TC_i^u,$$

$$\frac{dw_j^2}{dt} = 2w_j \frac{dw_j}{dt} + \frac{(dw_j)^2}{dt} = -2w_j \frac{\partial \ell}{\partial w_j} + TC_j^w, \tag{29}$$

where $C_i^u = \mathrm{Var}[\frac{\partial \ell}{\partial u_i}]$ and $C_j^w = \mathrm{Var}[\frac{\partial \ell}{\partial w_j}]$. With Eq. (26), we obtain

$$\frac{d}{dt}(\|u\|^2 - \|w\|^2) = -T(\sum_j C_j^w - \sum_i C_i^u) = -T\left(\sum_j \mathrm{Var}\left[\frac{\partial \ell}{\partial w_j}\right] - \sum_i \mathrm{Var}\left[\frac{\partial \ell}{\partial u_i}\right]\right). \tag{30}$$

Due to the rescaling symmetry, the loss function can be considered as a function of the matrix $uw^T$. Here we define a new loss function as $\tilde{\ell}(u_i w_j) = \ell(u_i, w_j)$. Hence, we have

$$\frac{\partial \ell}{\partial w_j} = \sum_i u_i \frac{\partial \tilde{\ell}}{\partial(u_i w_j)}, \frac{\partial \ell}{\partial u_i} = \sum_j w_j \frac{\partial \tilde{\ell}}{\partial(u_i w_j)}. \tag{31}$$

We can rewrite Eq. (30) into

$$\frac{d}{dt}(\|u\|^2 - \|w\|^2) = -T(u^T C_1 u - w^T C_2 w),, \tag{32}$$

where

$$(C_1)_{ij} = \mathbb{E}\left[\sum_k \frac{\partial \tilde{\ell}}{\partial(u_i w_k)} \frac{\partial \tilde{\ell}}{\partial(u_j w_k)}\right] - \sum_k \mathbb{E}\left[\frac{\partial \tilde{\ell}}{\partial(u_i w_k)}\right]\mathbb{E}\left[\frac{\partial \tilde{\ell}}{\partial(u_j w_k)}\right],$$

$$\equiv \mathbb{E}[A^T A] - \mathbb{E}[A^T]\mathbb{E}[A] \tag{33}$$

$$(C_2)_{kl} = \mathbb{E}\left[\sum_i \frac{\partial \tilde{\ell}}{\partial(u_i w_k)} \frac{\partial \tilde{\ell}}{\partial(u_i w_l)}\right] - \sum_i \mathbb{E}\left[\frac{\partial \tilde{\ell}}{\partial(u_i w_k)}\right]\mathbb{E}\left[\frac{\partial \tilde{\ell}}{\partial(u_i w_l)}\right]$$

$$\equiv \mathbb{E}[AA^T] - \mathbb{E}[A]\mathbb{E}[A^T], \tag{34}$$

where

$$(A)_{ik} \equiv \frac{\partial \tilde{\ell}}{\partial(u_i w_k)}. \tag{35}$$

The proof is thus complete. □

### B.3 PROOF OF THEOREM 2

*Proof.* This proof is based on the fact that if a certain condition is satisfied for all trajectories with probability 1, this condition is satisfied by the stationary distribution of the dynamics with probability 1.

Let us first consider the case of $D > 1$. We first show that any trajectory satisfies at least one of the following five conditions: for any $i$, (i) $v_i \to 0$, (ii) $L(\theta) \to 0$, or (iii) for any $k \neq l$, $(u_i^{(k)})^2 - (u_i^{(l)})^2 \to 0$.

The SDE for $u_i^{(k)}$ is

$$\frac{du_i^{(k)}}{dt} = -2\frac{v_i}{u_i^{(k)}}(\beta_1 v - \beta_2) + 2\frac{v_i}{u_i^{(k)}}\sqrt{\eta(\alpha_1 v^2 - 2\alpha_2 v + \alpha_3)}\frac{dW}{dt}, \tag{36}$$

where $v_i := \prod_{k=1}^D u_i^{(k)}$, and so $v = \sum_i v_i$. There exists rescaling symmetry between $u_i^{(k)}$ and $u_i^{(l)}$ for $k \neq l$. By the law of balance, we have

$$\frac{d}{dt}[(u_i^{(k)})^2 - (u_i^{(l)})^2] = -T[(u_i^{(k)})^2 - (u_i^{(l)})^2]\mathrm{Var}\left[\frac{\partial \ell}{\partial(u_i^{(k)} u_i^{(l)})}\right], \tag{37}$$

where

$$\mathrm{Var}\left[\frac{\partial \ell}{\partial (u_i^{(k)} u_i^{(l)})}\right] = (\frac{v_i}{u_i^{(k)} u_i^{(l)}})^2 (\alpha_1 v^2 - 2\alpha_2 v + \alpha_3) \tag{38}$$

with $v_i/(u_i^{(k)} u_i^{(l)}) = \prod_{s \neq k,l} u_i^{(s)}$. In the long-time limit, $(u_i^{(k)})^2$ converges to $(u_i^{(l)})^2$ unless $\mathrm{Var}\left[\frac{\partial \ell}{\partial (u_i^{(k)} u_i^{(l)})}\right] = 0$, which is equivalent to $v_i/(u_i^{(k)} u_i^{(l)}) = 0$ or $\alpha_1 v^2 - 2\alpha_2 v + \alpha_3 = 0$. These two conditions correspond to conditions (i) and (ii). The latter is because $\alpha_1 v^2 - 2\alpha_2 v + \alpha_3 = 0$ takes place if and only if $v = \alpha_2/\alpha_1$ and $\alpha_2^2 - \alpha_1 \alpha_3 = 0$ together with $L(\theta) = 0$. Therefore, at stationarity, we must have conditions (i), (ii), or (iii).

Now, we prove that when (iii) holds, the condition 2-(b) in the theorem statement must hold: for $D = 1$, $(\log|v_i| - \log|v_j|) = c_0$ with $\mathrm{sgn}(v_i) = \mathrm{sgn}(v_j)$. When (iii) holds, there are two situations. First, if $v_i = 0$, we have $u_i^{(}k) = 0$ for all $k$, and $v_i$ will stay $0$ for the rest of the trajectory, which corresponds to condition (i).

If $v_i \neq 0$, we have $u_i^{(k)} \neq 0$ for all $k$. Therefore, the dynamics of $v_i$ is

$$\frac{dv_i}{dt} = -2\sum_k \left(\frac{v_i}{u_i^{(k)}}\right)^2 (\beta_1 v - \beta_2) + 2\sum_k \left(\frac{v_i}{u_i^{(k)}}\right)^2 \sqrt{\eta(\alpha_1 v^2 - 2\alpha_2 v + \alpha_3)} \frac{dW}{dt} + 4\sum_{k,l} \left(\frac{v_i^3}{(u_i^{(k)} u_i^{(l)})^2}\right) \eta(\alpha_1 v^2 - 2\alpha_2 v + \alpha_3). \tag{39}$$

Comparing the dynamics of $v_i$ and $v_j$ for $i \neq j$, we obtain

$$\frac{dv_i/dt}{\sum_k (v_i/u_i^{(k)})^2} - \frac{dv_j/dt}{\sum_k (v_j/u_j^{(k)})^2} = 4\left(\frac{\sum_{m,l} v_i^3/(u_i^{(m)} u_i^{(l)})^2}{\sum_k (v_i/u_i^{(k)})^2} - \frac{\sum_{m,l} v_j^3/(u_j^{(m)} u_j^{(l)})^2}{\sum_k (v_j/u_j^{(k)})^2}\right) \eta(\alpha_1 v^2 - 2\alpha_2 v + \alpha_3)$$

$$= 4\left(v_i \frac{\sum_{m,l} v_i^2/(u_i^{(m)} u_i^{(l)})^2}{\sum_k (v_i/u_i^{(k)})^2} - v_j \frac{\sum_{m,l} v_j^2/(u_j^{(m)} u_j^{(l)})^2}{\sum_k (v_j/u_j^{(k)})^2}\right) \eta(\alpha_1 v^2 - 2\alpha_2 v + \alpha_3). \tag{40}$$

By condition (iii), we have $|u_i^{(0)}| = \cdots = |u_i^{(D)}|$, i.e., $(v_i/u_i^{(k)})^2 = (v_i^2)^{D/(D+1)}$ and $(v_i/u_i^{(m)} u_i^{(l)})^2 = (v_i^2)^{(D-1)/(D+1)}$.[8] Therefore, we obtain

$$\frac{dv_i/dt}{(D+1)(v_i^2)^{D/(D+1)}} - \frac{dv_j/dt}{(D+1)(v_j^2)^{D/(D+1)}} = \left(v_i \frac{D(v_i^2)^{(D-1)/(D+1)}}{2(v_i^2)^{D/(D+1)}} - v_j \frac{D(v_j^2)^{(D-1)/(D+1)}}{2(v_j^2)^{D/(D+1)}}\right) \eta(\alpha_1 v^2 - 2\alpha_2 v + \alpha_3). \tag{41}$$

We first consider the case where $v_i$ and $v_j$ initially share the same sign (both positive or both negative). When $D > 1$, the left-hand side of Eq. (41) can be written as

$$\frac{1}{1-D} \frac{dv_i^{2/(D+1)-1}}{dt} + 4Dv_i^{1-2/(D+1)} \eta(\alpha_1 v^2 - 2\alpha_2 v + \alpha_3) - \frac{1}{1-D} \frac{dv_j^{2/(D+1)-1}}{dt} - 4Dv_j^{1-2/(D+1)} \eta(\alpha_1 v^2 - 2\alpha_2 v + \alpha_3), \tag{42}$$

which follows from Ito's lemma:

$$\frac{dv_i^{2/(D+1)-1}}{dt} = \left(\frac{2}{D+1} - 1\right) v_i^{2/(D+1)-2} \frac{dv_i}{dt} + 2(\frac{2}{D+1} - 1)(\frac{2}{D+1} - 2) v_i^{2/(D+1)-3} \left(\sum_k (\frac{v_i}{u_i^{(k)}})^2 \sqrt{\eta(\alpha_1 v^2 - 2\alpha_2 v + \alpha_3)}\right)^2$$

$$= (\frac{2}{D+1} - 1) v_i^{2/(D+1)-2} \frac{dv_i}{dt} + 4D(D-1) v_i^{1-2/(D+1)} \eta(\alpha_1 v^2 - 2\alpha_2 v + \alpha_3). \tag{43}$$

Substitute in Eq. (41), we obtain Eq. (42).

Now, we consider the right-hand side of Eq. (41), which is given by

$$2Dv_i^{1-2/(D+1)} \eta(\alpha_1 v^2 - 2\alpha_2 v + \alpha_3) - 2Dv_j^{1-2/(D+1)} \eta(\alpha_1 v^2 - 2\alpha_2 v + \alpha_3). \tag{44}$$

---

[8]Here, we only consider the root on the positive real axis.

Combining Eq. (42) and Eq. (44), we obtain

$$\frac{1}{1-D}\frac{dv_i^{2/(D+1)-1}}{dt} - \frac{1}{1-D}\frac{dv_j^{2/(D+1)-1}}{dt} = -2D(v_i^{1-2/(D+1)} - v_j^{1-2/(D+1)})\eta(\alpha_1 v^2 - 2\alpha_2 v + \alpha_3).$$
(45)

By defining $z_i = v_i^{2/(D+1)-1}$, we can further simplify the dynamics:

$$\frac{d(z_i - z_j)}{dt} = 2D(D-1)\left(\frac{1}{z_i} - \frac{1}{z_j}\right)\eta(\alpha_1 v^2 - 2\alpha_2 v + \alpha_3)$$

$$= -2D(D-1)\frac{z_i - z_j}{z_i z_j}\eta(\alpha_1 v^2 - 2\alpha_2 v + \alpha_3).$$
(46)

Hence,

$$z_i(t) - z_j(t) = \exp\left[-\int dt \frac{2D(D-1)}{z_i z_j}\eta(\alpha_1 v^2 - 2\alpha_2 v + \alpha_3)\right].$$
(47)

Therefore, if $v_i$ and $v_j$ initially have the same sign, they will decay to the same value in the long-time limit $t \to \infty$, which gives condition 2-(b). When $v_i$ and $v_j$ initially have different signs, we can write Eq. (41) as

$$\frac{d|v_i|/dt}{(D+1)(|v_i|^2)^{D/(D+1)}} + \frac{d|v_j|/dt}{(D+1)(|v_j|^2)^{D/(D+1)}} = \left(|v_i|\frac{D(|v_i|^2)^{(D-1)/(D+1)}}{2(|v_i|^2)^{D/(D+1)}} + |v_j|\frac{D(|v_j|^2)^{(D-1)/(D+1)}}{2(|v_j|^2)^{D/(D+1)}}\right)$$
$$\times \eta(\alpha_1 v^2 - 2\alpha_2 v + \alpha_3).$$
(48)

Hence, when $D > 1$, we simplify the equation with a similar procedure as

$$\frac{1}{1-D}\frac{d|v_i|^{2/(D+1)-1}}{dt} + \frac{1}{1-D}\frac{d|v_j|^{2/(D+1)-1}}{dt} = -2D(|v_i|^{1-2/(D+1)} + |v_j|^{1-2/(D+1)})\eta(\alpha_1 v^2 - 2\alpha_2 v + \alpha_3).$$
(49)

Defining $z_i = |v_i|^{2/(D+1)-1}$, we obtain

$$\frac{d(z_i + z_j)}{dt} = 2D(D-1)\left(\frac{1}{z_i} + \frac{1}{z_j}\right)\eta(\alpha_1 v^2 - 2\alpha_2 v + \alpha_3)$$

$$= 2D(D-1)\frac{z_i + z_j}{z_i z_j}\eta(\alpha_1 v^2 - 2\alpha_2 v + \alpha_3),$$
(50)

which implies

$$z_i(t) + z_j(t) = \exp\left[\int dt \frac{2D(D-1)}{z_i z_j}\eta(\alpha_1 v^2 - 2\alpha_2 v + \alpha_3)\right].$$
(51)

From this equation, we reach the conclusion that if $v_i$ and $v_j$ have different signs initially, one of them converges to 0 in the long-time limit $t \to \infty$, corresponding to condition 1 in the theorem statement. Hence, for $D > 1$, at least one of the conditions is always satisfied at $t \to \infty$.

Now, we prove the theorem for $D = 1$, which is similar to the proof above. The law of balance gives

$$\frac{d}{dt}[(u_i^{(1)})^2 - (u_i^{(2)})^2] = -T[(u_i^{(1)})^2 - (u_i^{(2)})^2]\mathrm{Var}\left[\frac{\partial \ell}{\partial(u_i^{(1)} u_i^{(2)})}\right].$$
(52)

We can see that $|u_i^{(1)}| \to |u_i^{(2)}|$ takes place unless $\mathrm{Var}\left[\frac{\partial \ell}{\partial(u_i^{(1)} u_i^{(2)})}\right] = 0$, which is equivalent to $L(\theta) = 0$. This corresponds to condition (ii). Hence, if condition (ii) is violated, we need to prove condition (iii). In this sense, $|u_i^{(1)}| \to |u_i^{(2)}|$ occurs and Eq. (41) can be rewritten as

$$\frac{dv_i/dt}{|v_i|} - \frac{dv_j/dt}{|v_j|} = (\mathrm{sign}(v_i) - \mathrm{sign}(v_j))\eta(\alpha_1 v^2 - 2\alpha_2 v + \alpha_3).$$
(53)

When $v_i$ and $v_j$ are both positive, we have

$$\frac{dv_i/dt}{v_i} - \frac{dv_j/dt}{v_j} = 0.$$
(54)

With Ito's lemma, we have

$$\frac{d\log(v_i)}{dt} = \frac{dv_i}{v_i dt} - 2\eta(\alpha_1 v^2 - 2\alpha_2 v + \alpha_3).\tag{55}$$

Therefore, Eq. (54) can be simplified to

$$\frac{d(\log(v_i) - \log(v_j))}{dt} = 0,\tag{56}$$

which indicates that all $v_i$ with the same sign will decay at the same rate. This differs from the case of $D > 2$ where all $v_i$ decay to the same value. Similarly, we can prove the case where $v_i$ and $v_j$ are both negative.

Now, we consider the case where $v_i$ is positive while $v_j$ is negative and rewrite Eq. (53) as

$$\frac{dv_i/dt}{v_i} + \frac{d(|v_j|)/dt}{|v_j|} = 2\eta(\alpha_1 v^2 - 2\alpha_2 v + \alpha_3).\tag{57}$$

Furthermore, we can derive the dynamics of $v_j$ with Ito's lemma:

$$\frac{d\log(|v_j|)}{dt} = \frac{dv_i}{v_i dt} - 2\eta(\alpha_1 v^2 - 2\alpha_2 v + \alpha_3).\tag{58}$$

Therefore, Eq. (57) takes the form of

$$\frac{d(\log(v_i) + \log(|v_j|))}{dt} = -2\eta(\alpha_1 v^2 - 2\alpha_2 v + \alpha_3).\tag{59}$$

In the long-time limit, we can see $\log(v_i|v_j|)$ decays to $-\infty$, indicating that either $v_i$ or $v_j$ will decay to 0. This corresponds to condition 1 in the theorem statement. Combining Eq. (56) and Eq. (59), we conclude that all $v_i$ have the same sign as $t \to \infty$, which indicates condition 2-(a) if conditions in item 1 are all violated. The proof is thus complete. □

## B.4 STATIONARY DISTRIBUTION IN EQ. (13)

Following Eq. (39), we substitute $u_i^{(k)}$ with $v_i^{1/D}$ for arbitrary $k$ and obtain

$$\frac{dv_i}{dt} = -2(D+1)|v_i|^{2D/(D+1)}(\beta_1 v - \beta_2) + 2(D+1)|v_i|^{2D/(D+1)}\sqrt{\eta(\alpha_1 v^2 - 2\alpha_2 v + \alpha_3)}\frac{dW}{dt}$$
$$+ 2(D+1)Dv_i^3|v_i|^{-4/(D+1)}\eta(\alpha_1 v^2 - 2\alpha_2 v + \alpha_3).\tag{60}$$

With Eq. (47), we can see that for arbitrary $i$ and $j$, $v_i$ will converge to $v_j$ in the long-time limit. In this case, we have $v = dv_i$ for each $i$. Then, the SDE for $v$ can be written as

$$\frac{dv}{dt} = -2(D+1)d^{2/(D+1)-1}|v|^{2D/(D+1)}(\beta_1 v - \beta_2) + 2(D+1)d^{2/(D+1)-1}|v|^{2D/(D+1)}\sqrt{\eta(\alpha_1 v^2 - 2\alpha_2 v + \alpha_3)}\frac{dW}{dt}$$
$$+ 2(D+1)Dd^{4/(D+1)-2}v^3|v|^{-4/(D+1)}\eta(\alpha_1 v^2 - 2\alpha_2 v + \alpha_3).\tag{61}$$

If $v > 0$, Eq. (61) becomes

$$\frac{dv}{dt} = -2(D+1)d^{2/(D+1)-1}v^{2D/(D+1)}(\beta_1 v - \beta_2) + 2(D+1)d^{2/(D+1)-1}v^{2D/(D+1)}\sqrt{\eta(\alpha_1 v^2 - 2\alpha_2 v + \alpha_3)}\frac{dW}{dt}$$
$$+ 2(D+1)Dd^{4/(D+1)-2}v^{3-4/(D+1)}\eta(\alpha_1 v^2 - 2\alpha_2 v + \alpha_3).\tag{62}$$

Therefore, the stationary distribution of a general deep diagonal network is given by

$$p(v) \propto \frac{1}{v^{3(1-1/(D+1))}(\alpha_1 v^2 - 2\alpha_2 v + \alpha_3)}\exp\left(-\frac{1}{T}\int dv \frac{d^{1-2/(D+1)}(\beta_1 v - \beta_2)}{(D+1)v^{2D/(D+1)}(\alpha_1 v^2 - 2\alpha_2 v + \alpha_3)}\right).\tag{63}$$

If $v < 0$, Eq. (61) becomes

$$\frac{d|v|}{dt} = -2(D+1)d^{2/(D+1)-1}|v|^{2D/(D+1)}(\beta_1|v| + \beta_2) - 2(D+1)d^{2/(D+1)-1}|v|^{2D/(D+1)}\sqrt{\eta(\alpha_1|v|^2 + 2\alpha_2|v| + \alpha_3)}\frac{dW}{dt}$$
$$+ 2(D+1)Dd^{4/(D+1)-2}|v|^{3-4/(D+1)}\eta(\alpha_1|v|^2 + 2\alpha_2|v| + \alpha_3).\tag{64}$$

The stationary distribution of $|v|$ is given by

$$p(|v|) \propto \frac{1}{|v|^{3(1-1/(D+1))}(\alpha_1|v|^2 + 2\alpha_2|v| + \alpha_3)} \exp\left(-\frac{1}{T}\int d|v| \frac{d^{1-2/(D+1)}(\beta_1|v| + \beta_2)}{(D+1)|v|^{2D/(D+1)}(\alpha_1|v|^2 + 2\alpha_2|v| + \alpha_3)}\right). \tag{65}$$

Thus, we have obtained

$$p_\pm(|v|) \propto \frac{1}{|v|^{3(1-1/(D+1))}(\alpha_1|v|^2 \mp 2\alpha_2|v| + \alpha_3)} \exp\left(-\frac{1}{T}\int d|v| \frac{d^{1-2/(D+1)}(\beta_1|v| \mp \beta_2)}{(D+1)|v|^{2D/(D+1)}(\alpha_1|v|^2 \mp 2\alpha_2|v| + \alpha_3)}\right). \tag{66}$$

Especially when $D = 1$, the distribution function can be simplified as

$$p_\pm(|v|) \propto \frac{|v|^{\pm\beta_2/2\alpha_3 T - 3/2}}{(\alpha_1|v|^2 \mp 2\alpha_2|v| + \alpha_3)^{1\pm\beta_2/4T\alpha_3}} \exp\left(-\frac{1}{2T}\frac{\alpha_3\beta_1 - \alpha_2\beta_2}{\alpha_3\sqrt{\Delta}}\arctan\frac{\alpha_1|v| \mp \alpha_2}{\sqrt{\Delta}}\right), \tag{67}$$

where we have used the integral

$$\int dv \frac{\beta_1 v \mp \beta_2}{\alpha_1 v^2 - 2\alpha_2 v + \alpha_3} = \frac{\alpha_3\beta_1 - \alpha_2\beta_2}{\alpha_3\sqrt{\Delta}}\arctan\frac{\alpha_1|v| \mp \alpha_2}{\sqrt{\Delta}} \pm \frac{\beta_2}{\alpha_3}\log(v) \pm \frac{\beta_2}{2\alpha_3}\log(\alpha_1 v^2 - 2\alpha_2 v + \alpha_3). \tag{68}$$

## B.5  ANALYSIS OF THE MAXIMUM PROBABILITY POINT

To investigate the existence of the maximum point given in Eq. (16), we treat $T$ as a variable and study whether $(\beta_1 - 10\alpha_2 T)^2 + 28\alpha_1 T(\beta_2 - 3\alpha_3 T) := A$ in the square root is always positive or not. When $T < \frac{\beta_2}{3\alpha_3} = T_c/3$, $A$ is positive for arbitrary data. When $T > \frac{\beta_2}{3\alpha_3}$, we divide the discussion into several cases. First, when $\alpha_1\alpha_3 > \frac{25}{21}\alpha_2^2$, there always exists a root for the expression $A$. Hence, we find that

$$T = \frac{-5\alpha_2\beta_1 + 7\alpha_1\beta_2 + \sqrt{7}\sqrt{3\alpha_1\alpha_3\beta_1^2 - 10\alpha_1\alpha_2\beta_1\beta_2 + 7\alpha_1^2\beta_2^2}}{2(21\alpha_1\alpha_3 - 25\alpha_2^2)} := T^* \tag{69}$$

is a critical point. When $T_c/3 < T < T^*$, there exists a solution to the maximum condition. When $T > T^*$, there is no solution to the maximum condition.

The second case is $\alpha_2^2 < \alpha_1\alpha_3 < \frac{25}{21}\alpha_2^2$. In this case, we need to further compare the value between $5\alpha_2\beta_1$ and $7\alpha_1\beta_2$. If $5\alpha_2\beta_1 < 7\alpha_1\beta_2$, we have $A > 0$, which indicates that the maximum point exists. If $5\alpha_2\beta_1 > 7\alpha_1\beta_2$, we need to further check the value of minimum of $A$, which takes the form of

$$\min_T A(T) = \frac{(25\alpha_2^2 - 21\alpha_1\alpha_3)\beta_1^2 - (7\alpha_1\beta_2 - 5\alpha_2\beta_1)^2}{25\alpha_2^2 - 21\alpha_1\alpha_3}. \tag{70}$$

If $\frac{7\alpha_1}{5\alpha_2} < \frac{\beta_1}{\beta_2} < \frac{5\alpha_2 + \sqrt{25\alpha_2^2 - 21\alpha_1\alpha_3}}{3\alpha_3}$, the minimum of $A$ is always positive and the maximum exists. However, if $\frac{\beta_1}{\beta_2} \geq \frac{5\alpha_2 + \sqrt{25\alpha_2^2 - 21\alpha_1\alpha_3}}{3\alpha_3}$, there is always a critical learning rate $T^*$. If $\frac{\beta_1}{\beta_2} = \frac{5\alpha_2 + \sqrt{25\alpha_2^2 - 21\alpha_1\alpha_3}}{3\alpha_3}$, there is only one critical learning rate as $T_c = \frac{5\alpha_2\beta_1 - 7\alpha_1\beta_2}{2(25\alpha_2^2 - 21\alpha_1\alpha_3)}$. When $T_c/3 < T < T^*$, there is a solution to the maximum condition, while there is no solution when $T > T^*$. If $\frac{\beta_1}{\beta_2} > \frac{5\alpha_2 + \sqrt{25\alpha_2^2 - 21\alpha_1\alpha_3}}{3\alpha_3}$, there are two critical points:

$$T_{1,2} = \frac{-5\alpha_2\beta_1 + 7\alpha_1\beta_2 \mp \sqrt{7}\sqrt{3\alpha_1\alpha_3\beta_1^2 - 10\alpha_1\alpha_2\beta_1\beta_2 + 7\alpha_1^2\beta_2^2}}{2(21\alpha_1\alpha_3 - 25\alpha_2^2)}. \tag{71}$$

For $T < T_1$ and $T > T_2$, there exists a solution to the maximum condition. For $T_1 < T < T_2$, there is no solution to the maximum condition. The last case is $\alpha_2^2 = \alpha_1\alpha_3 < \frac{25}{21}\alpha_2^2$. In this sense, the expression of $A$ is simplified as $\beta_1^2 + 28\alpha_1\beta_2 T - 20\alpha_2\beta_1 T$. Hence, when $\frac{\beta_1}{\beta_2} < \frac{7\alpha_1}{5\alpha_2}$, there is no critical learning rate and the maximum always exists. Nevertheless, when $\frac{\beta_1}{\beta_2} > \frac{7\alpha_1}{5\alpha_2}$, there is always a critical learning rate as $T^* = \frac{\beta_1^2}{20\alpha_2\beta_1 - 28\alpha_1\beta_2}$. When $T < T^*$, there is a solution to the maximum condition, while there is no solution when $T > T^*$.

| | without weight decay | with weight decay |
|---|---|---|
| single layer | $(\alpha_1 v^2 - 2\alpha_2 v + \alpha_3)^{-1-\frac{\beta_1}{2T\alpha_1}}$ | $\alpha_1(v-k)^{-2-\frac{(\beta_1+\gamma)}{T\alpha_1}}$ |
| non-interpolation | $\dfrac{v^{\beta_2/2\alpha_3 T - 3/2}}{(\alpha_1 v^2 - 2\alpha_2 v + \alpha_3)^{1+\beta_2/4T\alpha_3}}$ | $\dfrac{v^{S(\beta_2-\gamma)/2\alpha_3\lambda - 3/2}}{(\alpha_1 v^2 - 2\alpha_2 v + \alpha_3)^{1+(\beta_2-\gamma)/4T\alpha_3}}$ |
| interpolation $y = kx$ | $\dfrac{v^{-3/2+\beta_1/2T\alpha_1 k}}{(v-k)^{2+\beta_1/2T\alpha_1 k}}$ | $\dfrac{v^{-3/2+\frac{1}{2T\alpha_1 k}(\beta_1-\frac{\gamma}{k})}}{(v-k)^{2+\frac{1}{2T\alpha_1 k}(\beta_1-\frac{\gamma}{k})}} \exp\!\left(-\dfrac{\beta\gamma}{2T\alpha_1}\dfrac{1}{k(k-v)}\right)$ |

Table 1: Summary of distributions $p(v)$ in a depth-1 neural network. Here, we show the distribution in the nontrivial subspace when the data $x$ and $y$ are positively correlated. The $\Theta(1)$ factors are neglected for concision.

## B.6 OTHER CASES FOR $D = 1$

The other cases are worth studying. For the interpolation case where the data is linear ($y = kx$ for some $k$), the stationary distribution is different and simpler. There exists a nontrivial fixed point for $\sum_i (u_i^2 - w_i^2)$: $\sum_j u_j w_j = \frac{\alpha_2}{\alpha_1}$, which is the global minimizer of $L$ and also has a vanishing noise. It is helpful to note the following relationships for the data distribution when it is linear:

$$\begin{cases} \alpha_1 = \mathrm{Var}[x^2], \\ \alpha_2 = k\mathrm{Var}[x^2] = k\alpha_1, \\ \alpha_3 = k^2\alpha_1, \\ \beta_1 = \mathbb{E}[x^2], \\ \beta_2 = k\mathbb{E}[x^2] = k\beta_1. \end{cases} \tag{72}$$

Since the analysis of the Fokker-Planck equation is the same, we directly begin with the distribution function in Eq. (14) for $u_i = -w_i$ which is given by $P(|v|) \propto \delta(|v|)$. Namely, the only possible weights are $u_i = w_i = 0$, the same as the non-interpolation case. This is because the corresponding stationary distribution is

$$P(|v|) \propto \frac{1}{|v|^2(|v|+k)^2} \exp\left(-\frac{1}{2T}\int d|v|\frac{\beta_1(|v|+k) + \alpha_1\frac{1}{T}(|v|+k)^2}{\alpha_1|v|(|v|+k)^2}\right)$$
$$\propto |v|^{-\frac{3}{2}-\frac{\beta_1}{2T\alpha_1 k}}(|v|+k)^{-2+\frac{\beta_1}{2T\alpha_1 k}}. \tag{73}$$

The integral of Eq. (73) with respect to $|v|$ diverges at the origin due to the factor $|v|^{\frac{3}{2}+\frac{\beta_1}{2T\alpha_1 k}}$.

For the case $u_i = w_i$, the stationary distribution is given from Eq. (14) as

$$P(v) \propto \frac{1}{v^2(v-k)^2} \exp\left(-\frac{1}{2T}\int dv\frac{\beta_1(v-k) + \alpha_1 T(v-k)^2}{\alpha_1 v(v-k)^2}\right)$$
$$\propto v^{-\frac{3}{2}+\frac{\beta_1}{2T\alpha_1 k}}(v-k)^{-2-\frac{\beta_1}{2T\alpha_1 k}}. \tag{74}$$

Now, we consider the case of $\gamma \neq 0$. In the non-interpolation regime, when $u_i = -w_i$, the stationary distribution is still $p(v) = \delta(v)$. For the case of $u_i = w_i$, the stationary distribution is the same as in Eq. (14) after replacing $\beta$ with $\beta_2' = \beta_2 - \gamma$. It still has a phase transition. The weight decay has the effect of shifting $\beta_2$ by $-\gamma$. In the interpolation regime, the stationary distribution is still $p(v) = \delta(v)$ when $u_i = -w_i$. However, when $u_i = w_i$, the phase transition still exists since the stationary distribution is

$$p(v) \propto \frac{v^{-\frac{3}{2}+\theta_2}}{(v-k)^{2+\theta_2}} \exp\left(-\frac{\beta_1\gamma}{2T\alpha_1}\frac{1}{k(k-v)}\right), \tag{75}$$

where $\theta_2 = \frac{1}{2T\alpha_1 k}(\beta_1 - \frac{\gamma}{k})$. The phase transition point is $\theta_2 = 1/2$, which is the same as the non-interpolation one.

The last situation is rather special, which happens when $\Delta = 0$ but $y \neq kx$: $y = kx - c/x$ for some $c \neq 0$. In this case, the parameters $\alpha$ and $\beta$ are the same as those given in Eq. (72) except for $\beta_2$:

$$\beta_2 = k\mathbb{E}[x^2] - kc = k\beta_1 - kc. \tag{76}$$

The corresponding stationary distribution is

$$P(|v|) \propto \frac{|v|^{-\frac{3}{2}-\phi_2}}{(|v|+k)^{2-\phi_2}} \exp\left(\frac{c}{2T\alpha_1} \frac{1}{k(k+|v|)}\right), \tag{77}$$

where $\phi_2 = \frac{1}{2T\alpha_1 k}(\beta_1 - c)$. Here, we see that the behavior of stationary distribution $P(|v|)$ is influenced by the sign of $c$. When $c < 0$, the integral of $P(|v|)$ diverges due to the factor $|v|^{-\frac{3}{2}-\phi_2} < |v|^{-3/2}$ and Eq. (77) becomes $\delta(|v|)$ again. However, when $c > 0$, the integral of $|v|$ may not diverge. The critical point is $\frac{3}{2} + \phi_2 = 1$ or equivalently: $c = \beta_1 + T\alpha_1 k$. This is because when $c < 0$, the data points are all distributed above the line $y = kx$. Hence, $u_i = -w_i$ can only give a trivial solution. However, if $c > 0$, there is the possibility to learn the negative slope $k$. When $0 < c < \beta_1 + T\alpha_1 k$, the integral of $P(|v|)$ still diverges and the distribution is equivalent to $\delta(|v|)$. Now, we consider the case of $u_i = w_i$. The stationary distribution is

$$P(|v|) \propto \frac{|v|^{-\frac{3}{2}+\phi_2}}{(|v|-k)^{2+\phi_2}} \exp\left(-\frac{c}{2T\alpha_1} \frac{1}{k-|v|}\right). \tag{78}$$

It also contains a critical point: $-\frac{3}{2} + \phi_2 = -1$, or equivalently, $c = \beta_1 - \alpha_1 kT$. There are two cases. When $c < 0$, the probability density only has support for $|v| > k$ since the gradient always pulls the parameter $|v|$ to the region $|v| > k$. Hence, the divergence at $|v| = 0$ is of no effect. When $c > 0$, the probability density has support on $0 < |v| < k$ for the same reason. Therefore, if $\beta_1 > \alpha_1 kT$, there exists a critical point $c = \beta_1 - \alpha_1 kT$. When $c > \beta_1 - \alpha_1 kT$, the distribution function $P(|v|)$ becomes $\delta(|v|)$. When $c < \beta_1 - \alpha_1 kT$, the integral of the distribution function is finite for $0 < |v| < k$, indicating the learning of the neural network. If $\beta_1 \le \alpha_1 kT$, there will be no criticality and $P(|v|)$ is always equivalent to $\delta(|v|)$. The effect of having weight decay can be similarly analyzed, and the result can be systematically obtained if we replace $\beta_1$ with $\beta_1 + \gamma/k$ for the case $u_i = -w_i$ or replacing $\beta_1$ with $\beta_1 - \gamma/k$ for the case $u_i = w_i$.

### B.7 SECOND-ORDER LAW OF BALANCE

Considering the modified loss function:

$$\ell_{\text{tot}} = \ell + \frac{1}{4}T\|\nabla L\|^2. \tag{79}$$

In this case, the Langevin equations become

$$dw_j = -\frac{\partial \ell}{\partial w_j}dt - \frac{1}{4}T\frac{\partial \|\nabla L\|^2}{\partial w_j}, \tag{80}$$

$$du_i = --\frac{\partial \ell}{\partial u_i}dt - \frac{1}{4}T\frac{\partial \|\nabla L\|^2}{\partial u_i}. \tag{81}$$

Hence, the modified SDEs of $u_i^2$ and $w_j^2$ can be rewritten as

$$\frac{du_i^2}{dt} = 2u_i\frac{du_i}{dt} + \frac{(du_i)^2}{dt} = -2u_i\frac{\partial \ell}{\partial u_i} + +TC_i^u - \frac{1}{2}Tu_i\nabla_{u_i}|\nabla L|^2, \tag{82}$$

$$\frac{dw_j^2}{dt} = 2w_j\frac{dw_j}{dt} + \frac{(dw_j)^2}{dt} = -2w_j\frac{\partial \ell}{\partial w_j} + TC_j^w - \frac{1}{2}Tw_j\nabla_{w_j}|\nabla L|^2. \tag{83}$$

In this section, we consider the effects brought by the last term in Eqs. (82) and (83). From the infinitesimal transformation of the rescaling symmetry:

$$\sum_j w_j\frac{\partial \ell}{\partial w_j} = \sum_i u_i\frac{\partial \ell}{\partial u_i}, \tag{84}$$

we take the derivative of both sides of the equation and obtain

$$\frac{\partial L}{\partial u_i} + \sum_j u_j\frac{\partial^2 L}{\partial u_i\partial u_j} = \sum_j w_j\frac{\partial^2 L}{\partial u_i\partial w_j}, \tag{85}$$

$$\sum_j u_j\frac{\partial^2 L}{\partial w_i\partial u_j} = \frac{\partial L}{\partial w_i} + \sum_j w_j\frac{\partial^2 L}{\partial w_i\partial w_j}, \tag{86}$$

where we take the expectation to $\ell$ at the same time. By substituting these equations into Eqs. (82) and (83), we obtain

$$\frac{d\|u\|^2}{dt} - \frac{d\|w\|\|^2}{dt} = T\sum_i (C_i^u + (\nabla_{u_i} L)^2) - T\sum_j (C_j^w + (\nabla_{w_j} L)^2). \tag{87}$$

Then following the procedure in Appendix. B.2, we can rewrite Eq. (87) as

$$\frac{d\|u\|^2}{dt} - \frac{d\|w\|^2}{dt} = -T(u^T C_1 u + u^T D_1 u - w^T C_2 w - w^T D_2 w)$$
$$= -T(u^T E_1 u - w^T E_2 w), \tag{88}$$

where

$$(D_1)_{ij} = \sum_k \mathbb{E}\left[\frac{\partial \ell}{\partial (u_i w_k)}\right] \mathbb{E}\left[\frac{\partial \ell}{\partial (u_j w_k)}\right], \tag{89}$$

$$(D_2)_{kl} = \sum_i \mathbb{E}\left[\frac{\partial \ell}{\partial (u_i w_k)}\right] \mathbb{E}\left[\frac{\partial \ell}{\partial (u_i w_l)}\right], \tag{90}$$

$$(E_1)_{ij} = \mathbb{E}\left[\sum_k \frac{\partial \ell}{\partial (u_i w_k)} \frac{\partial \ell}{\partial (u_j w_k)}\right], \tag{91}$$

$$(E_2)_{kl} = \mathbb{E}\left[\sum_i \frac{\partial \ell}{\partial (u_i w_k)} \frac{\partial \ell}{\partial (u_i w_l)}\right]. \tag{92}$$

For one-dimensional parameters $u, w$, Eq. (88) is reduced to

$$\frac{d}{dt}(u^2 - w^2) = -\mathbb{E}\left[\left(\frac{\partial \ell}{\partial (uw)}\right)^2\right](u^2 - w^2). \tag{93}$$

Therefore, we can see this loss modification increases the speed of convergence. Now, we move to the stationary distribution of the parameter $v$. At the stationarity, if $u_i = -w_i$, we also have the distribution $P(v) = \delta(v)$ like before. However, when $u_i = w_i$, we have

$$\frac{dv}{dt} = -4v(\beta_1 v - \beta_2) + 4Tv(\alpha_1 v^2 - 2\alpha_2 v + \alpha_3) - 4\beta_1^2 Tv(\beta_1 v - \beta_2)(3\beta_1 v - \beta_2) + 4v\sqrt{T(\alpha_1 v^2 - 2\alpha_2 v + \alpha_3)}\frac{dW}{dt}. \tag{94}$$

Hence, the stationary distribution becomes

$$P(v) \propto \frac{v^{\beta_2/2\alpha_3 T - 3/2 - \beta_2^2/2\alpha_3}}{(\alpha_1 v^2 - 2\alpha_2 v + \alpha_3)^{1+\beta_2/4T\alpha_3 + K_1}} \exp\left(-\left(\frac{1}{2T}\frac{\alpha_3\beta_1 - \alpha_2\beta_2}{\alpha_3\sqrt{\Delta}} + K_2\right)\arctan\frac{\alpha_1 v - \alpha_2}{\sqrt{\Delta}}\right), \tag{95}$$

where

$$K_1 = \frac{3\alpha_3\beta_1^2 - \alpha_1\beta_2^2}{4\alpha_1\alpha_3},$$
$$K_2 = \frac{3\alpha_2\alpha_3\beta_1^2 - 4\alpha_1\alpha_3\beta_1\beta_2 + \alpha_1\alpha_2\beta_2^2}{2\alpha_1\alpha_3\sqrt{\Delta}}. \tag{96}$$

From the expression above we can see $K_1 \ll 1 + \beta_2/4T\alpha_3$ and $K_2 \ll (\alpha_3\beta_1 - \alpha_2\beta_2)/2T\alpha_3\sqrt{\Delta}$. Hence, the effect of modification can only be seen in the term proportional to $v$. The phase transition point is modified as

$$T_c = \frac{\beta_2}{\alpha_3 + \beta_2^2}. \tag{97}$$

Compared with the previous result $T_c = \frac{\beta_2}{\alpha_3}$, we can see the effect of the loss modification is $\alpha_3 \to \alpha_3 + \beta_2^2$, or equivalently, $\mathrm{Var}[xy] \to \mathbb{E}[x^2 y^2]$. This effect can be seen from $E_1$ and $E_2$.

