# OpenReview forum: "Law of Balance and Stationary Distribution of Stochastic Gradient Descent"
_ICLR.cc/2024/Conference — Submitted to ICLR 2024_

### Official Review · Reviewer_8S7u · 2023-10-27

**Soundness:** 2 fair
**Presentation:** 2 fair
**Contribution:** 2 fair
**Rating:** 3
**Confidence:** 4

**Summary:**

This work considers (mini-batch) SGD on loss functions with re-scaling symmetry. This work shows that SGD tends to find a solution with balanced weights. Furthermore, this work also analyzes the stationary distribution induced by a continuously approximated SGD for a diagonal linear network, where several complicated behaviors of SGD are discussed, such as phase transition, loss of ergodicity, and fluctuation inversion.

**Strengths:**

+ The focus of this work, that is, the behavior of SGD, is an important and relevant topic to the community.
+ Some of the findings on the differences between SGD and (noisy) GD might be interesting.
+ The calculation of the stationary distribution of SGD for diagonal linear networks should be new to my knowledge.

**Weaknesses:**

- Some statements/writing are not precise and might be misleading. See more in the Question section.
- Some messages are already known from prior papers. For example, it is known (references are actually mentioned in this work) that SGD should not be approximated by gradient flow or gradient Langevin dynamic and that the SGD noise is parameter dependent. This paper should have been more careful in terms of clarifying the contributions.
- Not sure how relevant is the theory in this paper to practice. See more in the Question section.

**Questions:**

1. Discussions before Theorem 1. "....For example, it appears in any neural network with the ReLU activation". This might not be true. For example, $(u \\max \\{ w x, 0 \\} -y)\^2$ is rescaling symmetric only for non-negative $\\lambda$.

2. Theorem 1 and the follow-up discussions. Note that $C\_1 $ and $C\_2$ are functions of $u$ and $v$. Therefore, $\\lambda\_{1m}, \\lambda\_{1M}, \\lambda\_{2m}, \\lambda\_{2M}$ are not constants but functions of $u$ and $v$. Hence, eqs (5) or (6) do not directly imply that $u$ and $v$ are approximately balanced. Could the authors make some clarifications?

3. Discussions after eq (8). There seems to be a typo in the display after "....can be upper-bounded by an exponentially
decreasing function in time:...". Please clarify.

In addition, in eq(7), the coefficient $\\alpha\_1 v\^2 - 2\\alpha\_2 v  + \\alpha\_3$ is not a constant but a function of $v$ (hence a function of $u$ and $w$). So the statement that "....can be upper-bounded by an exponentially decreasing function in time" might not be accurate.

4. Discussions after eq (9). There might be a typo in the definition of $C(v)$. It should be the variance of the gradient of the loss.

5. "These relations imply that $C$ can be quite independent of $L$, contrary to popular beliefs in the literature...". Here, should the word "independent" be revised to "dependent"?

6. The diagonal linear network studied in this paper takes a 1-dimnensional input. However, prior works consider diagonal linear network with a multivariate input. So the setting in this work might be limited compared to prior works.

Overall, I feel the theory in this work is not entirely precise. Plus, the model setup is limited, and the results might not be general and might be strongly rely on this particular setup.

---

> ### Author Response · Authors · 2023-11-21
> **Author reply part 1**
>
> Thank you for the detailed feedback. We answer both the weaknesses and questions below.
>
> Weaknesses:
>
> **1. Some statements/writing are not precise and might be misleading. See more in the Question section.**
>
> Thanks for this criticism. We have done our best to update the manuscript to improve the precision of our language. See our answers below.
>
> **2. Some messages are already known from prior papers. For example, it is known (references are actually mentioned in this work) that SGD should not be approximated by gradient flow or gradient Langevin dynamic and that the SGD noise is parameter dependent. This paper should have been more careful in terms of clarifying the contributions.**
>
> Thanks for this comment. It is not our intention to claim these messages as our contributions. To be precise, the following are NOT the contributions of this work:
> 1. SGD should not be approximated by gradient flow or gradient Langevin dynamics
> 2. SGD noise is parameter-dependent
>
> We have added references to attribute these contributions to previous works, which should remove this potential confusion from the audience. Also, please let us know if you are aware of relevant works that we did not discuss.
>
> That being said, the following are the primary and original contributions of our work:
> 1. The identification of the law of balance (along with its various implications, such as SGD having zero probability of accessing certain types of unbalanced solutions), which is a general property of the SGD algorithm, as the law only assumes the symmetry and local differentiability
> 2. The first analytical and exact calculation of the stationary distribution of SGD in ANY high-dimensional situation (along with many unknown phenomena we discovered with its solution, such as the fluctuation inversion, qualitative difference between networks of different depths, etc.). Prior to our work, there has not been any case where the invariant measure of SGD is precisely known.
>
> **3. Not sure how relevant is the theory in this paper to practice. See more in the Question section.**
>
> We have updated the manuscript to better clarify our contribution and its relevance to practice. As we discuss below, our theory certainly applies to ReLU activations. Also, the fact that the norms are balanced is only an implication of the law of balance. The law of balance states that there exists a quantity whose time-evolution obeys an ordinary differential equation, even if the dynamics of the parameters obey stochastic differential equations, which is an insightful mathematical result. A practically relevant implication is the balance. As we will show below, the balance indeed holds in rather practical situations.
>
>
> Questions:
>
> **1: Discussions before Theorem 1. "....For example, it appears in any neural network with the ReLU activation". This might not be true. For example $(u\max\{wx,0\}-y)^2$,  is rescaling symmetric only for non-negative $\lambda$.**
>
> Thanks for this comment. We are afraid to say that this is a misunderstanding of our result. Theorem 1 also holds for the rescaling symmetry in the positive real axis. In fact, Theorem 1 only requires the rescaling symmetry to hold in the vicinity of $\lambda = 1$. Namely, the only condition that is required to prove Theorem 1 is the infinitesimal transformation $\ell(u,w,x)=\ell((1+\epsilon)u,(1-\epsilon)w,x)$.
>
> Therefore, the law of balance applies to even broader situations, including the ReLU activation as a special case. We have updated the manuscript (above Theorem 1) to clarify this part.
>
> **2: Theorem 1 and the follow-up discussions. Note that $C_1$ and $C_2$ are functions of $u$ and $v$. Therefore, $\lambda_{1m},\lambda_{1M},\lambda_{2m},\lambda_{2M}$ are not constants but functions of $u$ and $v$. Hence, eqs (5) or (6) do not directly imply that $u$ and $v$ are approximately balanced. Could the authors make some clarifications?**
>
> This is a good and sharp question. It is true that $C_1$ and $C_2$ are functions of $u$ and $v$. However, at the end of the training, when the model stabilizes with a local minimum valley, $C_1$ and $C_2$ will be approximately stationary and time-independent. When this is the case, any discrepancy between $C_1$ and $C_2$ will drive the norm difference towards balance. To substantiate this argument, we have included an experiment to show that $C_1$ and $C_2$ indeed converge to stationary constants during the training of a ReLU neural network. See the new A1 section.

---

> ### Author Response · Authors · 2023-11-21
> **Author reply part 2**
>
> **3: Discussions after eq (8). There seems to be a typo in the display after "....can be upper-bounded by an exponentially decreasing function in time:...". Please clarify. In addition, in eq(7), the coefficient $\alpha_1v^2-2\alpha_2v+\alpha_3$ is not a constant but a function of $v$ (hence a function of $u$ and $w$). So the statement that "....can be upper-bounded by an exponentially decreasing function in time" might not be accurate.**
>
> Thanks for this question. This is not a typo.
>
> While the expression $\alpha_1v^2-2\alpha_2v+\alpha_3$ is a function of $v$, it can be lower bounded the decay rate in Equation (7): $\alpha_1v^2-2\alpha_2v+\alpha_3\geq(\alpha_1\alpha_3-\alpha_2^2)/\alpha_1$, where $\alpha_1,\alpha_2,\alpha_3$ are constants independent of $u$ and $w$. Inserting the inequality to Equation (7) shows that the relevant quantity "....can be upper-bounded by an exponentially decreasing function in time:..." There is no mistake.
>
> **4: Discussions after eq (9). There might be a typo in the definition of $C(v)$. It should be the variance of the gradient of the loss.**
>
> Thanks for pointing this out. This is indeed a typo. We have corrected this part to $C(v)=\text{Var}[\nabla_v\ell(v,x)]=\cdots$ in the updated manuscript.
>
> **5: "These relations imply that $C$ can be quite independent of $L$, contrary to popular beliefs in the literature...". Here, should the word "independent" be revised to "dependent"?**
>
> Thanks for this question. Here, we say "independent" because the noise only depends on the variance of the data $x^2$ and $xy$, which do not enter the loss function $L$. In general, the variance and the expectation of the same data are independent of each other.  Also, while both $C$ and $L$ are functions of $u$ and $w$, their functional dependencies are different and unrelated to each other. This tells us that the noise is determined by the higher-order structure of the data, which is contrary to previous arguments in Ref. [23,27].  Therefore, what we intended to say is that the training loss and the noise covariance are not functions of each other and need to be analyzed separately. We have clarified this part in the manuscript.
>
>
> **6: The diagonal linear network studied in this paper takes a 1-dimnensional input. However, prior works consider diagonal linear network with a multivariate input. So the setting in this work might be limited compared to prior works.**
>
> Thanks for raising this question. First of all, we would like to emphasize that our most significant contribution in the second part of the paper is NOT to derive the stationary distribution of deep diagonal networks and help us understand a diagonal network. Instead, the significance of this part is to analytically derive the stationary distribution of SGD in a high dimensional and highly nonconvex setting without any assumption-- something that has been a difficult open problem for many years and no previous work has achieved. For example, this point has been acknowledged by Reviewer egby and Y3Mm. Therefore, it is not fair to compare the problem setting of our work to previous works that specialize in studying diagonal networks. Instead, it is fair to compare our work with any work that derives the invariant measure of SGD, which is nonexistent. It is also fair to compare our work with previous works that study the stationary properties of SGD, and our result greatly advances our understanding of this problem because all stationary properties of SGD can be directly derived from the stationary distribution of SGD.
>
> That being said, for the diagonal linear network with multivariate input, the Law of Balance in Theorem 1 and Corollary 1 still holds due to the rescaling symmetry. This indicates that $u_i$ and $w_i$ still converge to the same value in the long-time limit. However, the discussion in the stationary distribution, i.e., Theorem 2, does not hold any longer since here $v_i$ are $v_j$ are always coupled to each other in the gradient of the loss function $L$ and the noise term in the Langevin equation. It is actually unlikely that there exists an analytical solution to the stationary distribution for the case of multivariate input.

---

> > ### Comment · Reviewer_8S7u · 2023-11-22
> > **Follow up questions**
> >
> > Thank you for your comments.
> >
> > Q2. Thank you for confirming that $C_1$ and $C_2$ are functions of $u$ and $v$. This is a major drawback of the theory. The significance of Section 2 is therefore limited and is much less than what has been claimed in the paper.
> >
> > Q3. I agree that one can do a lower bound, $\alpha_1 v^2 - 2\alpha_2 v + \alpha_3 \ge (\alpha_1 \alpha_3 - \alpha_2^2) / \alpha_1$. Would extra assumptions need to be made for the right-hand side of the inequality to be strictly positive?
> >
> > Q6. I still think the significance of the results is limited if only 1-dimensional data is allowed.
> >
> > Theorem 1 allows multi-variate inputs, but Theorem 1 has a major drawback as explained in Q2.
> >
> >  To summarize, I don't think this paper has made sufficient contributions.

---

> ### Author Response · Authors · 2023-11-23
> **Author Reply 2 Part 1**
>
> Thank you for your reply. We would like to further elaborate why our contribution is sufficient and valuable to our community.
>
> **Q2. Thank you for confirming that $C_1$ and $C_2$ are functions of $u$ and $v$. This is a major drawback of the theory. The significance of Section 2 is therefore limited and is much less than what has been claimed in the paper.**
>
> Thank you for the criticism. However, we politely disagree with the criticism. This is not a drawback of the theory but a truth and beauty of the theory. Our first major technical contribution is to show that the dynamics of the balance (difference between the two norms) follows Eq. (4), which holds true without any assumption. There is only truth and no drawback here.
>
> Regarding the significance of Section 2, we argue that the results here are both significant and useful, both when it implies norm balancing and when it does not. As our manuscript shows solidly, this significance holds both theoretically and empirically.
> Theoretically, when it does imply norm balancing, the norm balancing itself is an important implication. We believe that you agree to this part. So, let us focus on the case when it does not imply norm balancing. When it does not, it can also be useful. For example, in our results in Section 3, the stationary distribution for a special type of data (when the data is linear and has no noise in the label) can only be derived by using Eq. (4), even when it does not imply balance (see our answer to Q3 below). Without Eq. (4), one cannot derive the stationary distribution for SGD for this case.
>
> Empirically, in all experiments we conducted, the law of balance does lead to the balance condition. The reason is also clarified in our first round of reply. At the end of the training, $C_1$ and $C_2$ become approximately constants in time, and so their dependence on $u$ and $v$ no longer matters. This argument is also empirically supported by the new experiment we added in Section A1. Certainly, there will be corner cases where the law of balance does not imply norm balance, but are these cases of machine learning relevant? We believe this is an important open question. Also, our theory implies two more important open problems: can one construct an example where $C_1$ and $C_2$ do not become approximately stationary at the end of training? Also, can one construct meaningful and machine-learning examples where the law of balance does not imply norm balance? In fact, we believe that constructing these examples are important future problems of great scientific value and will significantly advance our understanding of SGD (because these examples will actually tell us when SGD will fail).
>
> Regarding "what has been claimed in the paper," we intend to ensure and are confident that all our claims are both accurate and precise. That being said, we invite you to be specific and point out specific claims that are not supported by the evidence given in the paper. We are happy to make further revisions if there is any remaining imprecision in the claims.
>
> Lastly, we believe it is important to realize that every theory has drawbacks and limitations. What really matters is (1) novelty of the insight and (2) its capability of explaining real phenomena that the theory aims to tackle. Our theory achieves both and significantly advances our understanding of SGD (also see our answer to Q6 below).
>
> **Q3. I agree that one can do a lower bound, $\alpha_1 v^2 - 2\alpha_2 v + \alpha_3 \ge (\alpha_1 \alpha_3 - \alpha_2^2) / \alpha_1$. Would extra assumptions need to be made for the right-hand side of the inequality to be strictly positive?**
>
> Thanks for the question. First of all, the quantity $(\alpha_1 \alpha_3 - \alpha_2^2) / \alpha_1$ is always non-negative: $\text{Var}[x^2]\text{Var}[y^2]\geq\text{Cov}[x^2,xy]^2$. Thus, the question is: when this quantity is zero? It happens only when the data takes the following special form: $y=kx +c/x$ for any constant $k$ and $c$. We also derived stationary distribution for this case (Section B6). Here, it is true that different layers are not guaranteed to be balanced, but one can still derive the stationary distribution of $v$. Crucially, in deriving its stationary distribution, one still needs the law of balance (namely, Eq. (4)). Thus, this is a great example showing how the law of balance is useful and significant even if the norm balance is not implied.
>
> Also, we would like to stress that our results in Section 3 do not rely on the positivity of $\alpha_1 \alpha_3 - \alpha_2^2$. We derived the stationary distribution for both $\alpha_1 \alpha_3 - \alpha_2^2>0$ and $\alpha_1 \alpha_3 - \alpha_2^2=0$. See Section B6 and Table 1 for their related discussion and results.

---

> ### Author Response · Authors · 2023-11-23
> **Author Reply 2 Part 2**
>
> **Q6. I still think the significance of the results is limited if only 1-dimensional data is allowed.**
>
> Thanks for this criticism. We agree that it is certainly a limitation that the stationary distribution we derived is only for 1-dimensional data. However, this is not a drawback but a reflection of our current most advanced understanding of the stationary distribution of SGD.
>
> That being said, we point out that this is a rather generic criticism, and it holds true that every theory has limited significance. The meaningful question is: the theory is limited in comparison to what? If any part of our message appears in previous works, please point it out, and we are happy to include a discussion and comparison with that work. We are confident that our theory does not follow trivially from any previous theory, and there is no easy way to derive anything more general/complicated than what the submitted manuscript has achieved. We also would like to ask two questions: is there a previous work that analytically derives the stationary of SGD for multivariate input? Or, can one prove a theorem showing that our result can be trivially derived from known results?
>
>
>
> **Theorem 1 allows multi-variate inputs, but Theorem 1 has a major drawback as explained in Q2. To summarize, I don't think this paper has made sufficient contributions.**
>
> Thanks for the criticism, which we cannot agree with. We politely ask you to review our answer above. To summarize, the significant contributions of our work are (1) pointing out the dynamics of SGD on the rescaling symmetry can be described by the law of balance, which is simple and strongly interpretable, and (2) providing the first analytical derivation of the stationary distribution of SGD. These are never proposed or studied in previous works. We also ask you to be more specific and constructive in the criticism. For example, it would be very helpful to point out specific claims that are not substantiated. It would also be less subjective to give explicit references and explain why, in light of these references, our contribution is not sufficient.

---

### Official Review · Reviewer_Y3Mm · 2023-10-31

**Soundness:** 4 excellent
**Presentation:** 4 excellent
**Contribution:** 4 excellent
**Rating:** 8
**Confidence:** 3

**Summary:**

The authors show that the noise of SGD from minibatching regularizes the solution towards a "balanced" solution whenever rescaling symmetries are present in the loss function. They then apply these results to derive the stationary distribution of SGD for a diagonal linear network. Then the authors characterize this stationary distribution, showing such phenomena as: phase transitions, loss of ergodicity, and fluctuation inversion. They show that these properties exist uniquely in deep networks, thus delineating a difference between deep and shallow networks.

**Strengths:**

- The paper provides new insights into the behavior of stochastic gradient descent (SGD), such as: how the Langevin model is flawed in studying SGD, that the noise in SGD creates a qualitative difference between it and gradient descent (GD), analysis between networks with and without depth, loss of ergodicity, among many others.
- Characterizing the stationary distribution of SGD analytically.
- Provided insights into why the Gibbs measure is bad for SGD.
- The paper is original, and very clearly written.

**Weaknesses:**

- The most complicated model they analyzed was a linear deep diagonal network, but I can imagine the general case being extremely difficult.

**Questions:**

- Can the properties found for SGD, such as "the qualitative difference between networks with different depths, the fluctuation inversion effect, the loss of ergodicity, and the incapability of learning a wrong sign for a deep model" be extended to loss functions without rescaling symmetries?

- How well do you believe the results in the paper transfer to nonlinear neural networks (i.e. with a nonlinear activation function)?

---

> ### Author Response · Authors · 2023-11-21
> **Author reply**
>
> Thanks for the encouraging review.
>
>
> **1. Can the properties found for SGD, such as "the qualitative difference between networks with different depths, the fluctuation inversion effect, the loss of ergodicity, and the incapability of learning a wrong sign for a deep model" be extended to loss functions without rescaling symmetries?**
>
> We do believe (and it is true) that these effects carry over to problems without rescaling symmetry, which is what the experiments on tanh nets exactly demonstrate (Figure 4 mid and right). Numerical results show that tanh nets (which do not have the rescaling symmetry) exhibit all four phenomena: the qualitative difference between networks with different depths, the fluctuation inversion effect, the loss of ergodicity, and the incapability of learning a wrong sign for a deep model. Therefore, this is a very good example of how solving an exactly solvable model leads to new insights into complicated problems that are not exactly solvable.
>
> In our opinion, an interesting and important remaining question is how to empirically measure these effects in more complicated models. For example, how to demonstrate that a network with more than a million parameters has the loss of ergodicity phenomenon? It is very difficult without a better theory because one cannot visualize such a landscape. One open problem is thus to identify a metric that correlates well with the loss of ergodicity phenomenon. The same problem applies to the "incapability of learning a wrong sign." For nonlinear models with non-monotonic activations, the meaning of this statement needs to be defined more clearly because the face value of this statement seems false (at least vague) for general problems. Thus, for these generally complicated models, there should be solutions that SGD has zero probability of accessing, but it is yet unclear what they are (while it should be a more nuanced version of "incapability of learning a wrong sign").
>
> **2. How well do you believe the results in the paper transfer to nonlinear neural networks (i.e. with a nonlinear activation function)?**
>
> There are two aspects to this question. The first is whether the phenomena we discovered appear in nonlinear models. It certainly does. As we have commented above, a lot of phenomena we discover do appear empirically in nonlinear networks.
>
> The second and deeper aspect of the question is: what part of deep learning is the diagonal linear network a good model of? We think there are certain aspects of deep learning that diagonal linear networks are good models of. For example, a two-layer diagonal network is the simplest model for which feature learning happens (in the sense that its NTK changes throughout training). Therefore, it could be a minimal model for studying the optimization of actual neural networks with a finite width. At the same time, there are aspects of deep learning that deep diagonal networks certainly fail to capture. The most obvious problem here is certainly that the output is a linear function of the input and so it does not seem a satisfactory model of the generalization power of deep learning. This is why we study the optimization dynamics of SGD in this problem but not its generalization power.

---

> > ### Comment · Reviewer_Y3Mm · 2023-11-22
> >
> > Thank you for your time and effort in the rebuttal. I have read the other reviews and author replies, as well as the revisions.

---

### Official Review · Reviewer_wHcL · 2023-11-03

**Soundness:** 3 good
**Presentation:** 3 good
**Contribution:** 2 fair
**Rating:** 5
**Confidence:** 2

**Summary:**

This paper proposed a law of balance phenomenon to better understand the SGD dynamics. Based on the proposed method, the authors have theoretically shown the unique properties separating the deep and shallow networks.

**Strengths:**

The theory is centered around the law of balance equation, which is clean and interpretable in some sense.

**Weaknesses:**

The boundness of the law of balance seems determined largely by the covariance matrices in equation 4. However, is there any guarantee on the condition of the matrices? what happens if the matrices are degenerate, and can you justify the matrices' degeneracy matter in practice?

**Questions:**

I would like to know more details about the case when the matrices are degenerate or justifications that they are non-degenerate.

**Details Of Ethics Concerns:**

no ethics concern

---

> ### Author Response · Authors · 2023-11-21
> **Author reply**
>
> Thank you for your comments and questions. We answer both the weakness and the question below.
>
> Weakness:
>
> **1. The boundness of the law of balance seems determined largely by the covariance matrices in equation 4. However, is there any guarantee on the condition of the matrices? what happens if the matrices are degenerate, and can you justify the matrices' degeneracy matter in practice?**
>
> This is a very good question. The balance condition does depend on the covariance of the gradient, and, in general, one cannot derive any guarantee on the form or structure of the gradient covariance -- this is a reflection of the generality of the law of balance as it only assumes having a symmetry in the loss function. Also, we note that the balancing of norms is just one implication of the law of balance. The law of balance that we have derived is nothing but Eq. (4), which directly states that the evolution of the norm difference during training depends on the gradient variance of two parts of the model, which already offers an important insight per se.
>
> That being said, if we restrict ourselves to the discussion of the noise covariance in deep learning, in some sense, the degeneracy is not quite a problem. See the answer below.
>
>
> Question:
>
> **1. I would like to know more details about the case when the matrices are degenerate or justifications that they are non-degenerate.
> This is also a good question. Please first see the answer above.**
>
> If we restrict ourselves to the discussion of the noise covariance in deep learning, the degeneracy is unlikely to be a problem for two reasons. First of all, at the end of the training, while many directions in the loss landscape have a small Hessian eigenvalue, it is usually not the case that these eigenvalues are zero (for example, see https://arxiv.org/abs/1706.04454 and https://arxiv.org/abs/2201.13011) -- and nonzero eigenvalues in the Hessian indicate a full-rank gradient noise -- it is possible that the eigenvalues of the covariance is very small, but they are rarely precisely zero.
>
> Now, a major reason (perhaps the only reason) for having zero eigenvalues in the gradient covariance or Hessian is when the loss function itself has a continuous symmetry, but this problem is alleviated in our theory because the two matrices in Eq. (4) are the gradient covariance of the surrogate loss function that has already removed the rescaling symmetries -- therefore if there is no other continuous symmetry in the loss function, it is quite unlikely that gradient covariance has precise zero eigenvalues throughout training.
>
> Lastly, even if there are degeneracies in the gradient covariance, these degeneracies need to precisely align with the directions of $u$ and $w$ for the balancing to fail. This is very unlikely in reality.
>
> To substantiate our argument, we have included a new experiment with a ReLU network in Section A1, where we plotted the first and second term in Eq. (4). In this experiment, it is clear that both terms are finite throughout training -- this implies that degeneracy is not really a problem in this case.

---

### Official Review · Reviewer_eqby · 2023-11-04

**Soundness:** 4 excellent
**Presentation:** 4 excellent
**Contribution:** 3 good
**Rating:** 6
**Confidence:** 4

**Summary:**

From the view of symmetry, this manuscript shows that SGD systematically moves towards a balanced solution when rescaling symmetry of loss function exists. The stationary distribution of model parameters of is also derived for simple diagonal linear networks. Many connections with other works also shown.

**Strengths:**

1. A novel view, symmery of loss, is provided for analyzing the solution of SGD and its stationary distribution.
2. The derived stationary distribution under different factors, including learning rate, batch size, data noise, model width and depth, is analytically derived. This result explains many interesting observed phenomenon in practice or colloborates with exsiting findings in other works.

**Weaknesses:**

1. The diagram of phase transition is a special case or general, which should have been clarified.
2. Given the symmetry view,  whether the diagonal linear network is a representative architecture for investigating SGD? Could the authors comment more regarding this?
3. Another concern is that when conducting the analysis, a L2-norm regularization term is added. How does this affect all the derived results and further interpretation?

**Questions:**

See the above

---

> ### Author Response · Authors · 2023-11-21
> **Author reply**
>
> Thank you for your comments and questions.
>
> **1. The diagram of phase transition is a special case or general, which should have been clarified.**
>
> The phase diagram is for a special case of the linear neural network where we set $y=kx+\epsilon$ with $\epsilon$ a Gaussian noise. which gives a representative example to illustrate how the input data influences the stationary distribution and characterizes the SGD. We have updated the manuscript to clarify this.
>
> **2. Given the symmetry view, whether the diagonal linear network is a representative architecture for investigating SGD? Could the authors comment more regarding this?**
>
> This is a very good question. We think there are certainly aspects of deep learning for which diagonal linear networks are good models. For example, a two-layer diagonal network is the simplest model for which feature learning happens (in the sense that its NTK changes throughout training). Therefore, it could be a minimal model for studying the optimization of actual finite-width neural networks with a finite depth. At the same time, there are aspects of deep learning that deep diagonal networks certainly fail to capture. The most obvious problem here is certainly that the output is a linear function of the input, and so it does not seem to be a satisfactory model of the generalization power of deep learning. This is why we study the optimization dynamics of SGD in this problem but not its generalization power.
>
>
> **3. Another concern is that when conducting the analysis, a L2-norm regularization term is added. How does this affect all the derived results and further interpretation?**
>
> Thanks for raising this question. As is clear from the context (for example, see the paragraph above Eq. (7)), our theory allows the L2 regularization term to be exactly zero, so the whole theory applies both with and without L2 regularization.
>
> How does the regularization term affect the result? According to our theory, the regularization term does something similar to the noise, which drives the quantity $u_i^2-w_i^2$ decay to zero exponentially, which is described by Eq. (7) in the manuscript. This term gives rise to a decay rate lower bounded by the coefficient $\gamma$. Furthermore, this regularization shifts the critical learning rate $T$ described by Eq. (15). If $\gamma>\beta_2$, there will be no phase transitions, and the parameters will flow to zero after training. Hence, here, the regularization term accelerates the parameters to balance and shifts the phase transition point. However, both the balance and the phase transition points still exist without regularization – this actually implies that the implicit regularization effect of SGD noise on the rescaling symmetry is almost equivalent to an explicit L2 regularization. We have added some discussion of this point below Corollary 1.

---

### Author Response · Authors · 2023-11-21
**Rebuttal summary**

First of all, we would like to thank all the reviewers for their careful and constructive feedback. We have carefully studied all the criticisms and feedback from the reviewers and adapted our manuscript accordingly to address the raised concerns. The updated parts are highlighted in orange.

With these revisions, we are confident that the manuscript has significantly been improved, and the contributions of our manuscript are now clearly presented.

To summarize, we have made the following revisions to the manuscript.
1. Revision of Theorem 1 to clarify why the result applies to the case of ReLU networks.
2. Addition of a new experiment (Section A1) to show that (1) the law of balance applies to ReLU nets and (2) the two terms in Eq. (4) converge to time-independent constants.
3. Discussion of how implicit bias is related to L2 regularization below Corollary 1.
4. Appropriate references are given to previous works that show that the SGD noise is parameter-dependent.
5. Typos and minor errors have been fixed.

---

### Meta-Review · Area_Chair_Qntr · 2023-12-10

**Metareview:**

This paper calculates the steady-state distribution of stochastic gradient descent (SGD) for loss functions with scale symmetry. The paper shows that in SGD finds "balanced solutions" where the norm of the parameters is about the same. The stationary distribution of SGD has been derived for 1-dimensional data. The paper discusses a number of interesting insights from this expression, e.g., fluctuation inversion. The authors have argued that this has not been calculated before. But this is not true, e.g., one can calculate the steady-state distribution of SGD for state-dependent noise and general losses and also specialize these calculations for quadratic objectives (https://arxiv.org/abs/1710.11029).

There was discussion with reviewers as to whether Theorem 1 is meaningful since the quantities C1 and C2 both depend upon the weights u, w. The authors have argued that these quantities are almost constant during training. But it is not true since these are the off-diagonal blocks of the empirical Fisher information matrix (FIM), which changes throughout training in general. It is also not true that these entries of the FIM become constant towards the end of the training (as was argued by the authors in the rebuttal). One can see the expressions in many papers, e.g., https://arxiv.org/abs/1910.05929. Theorem 1 on the law of balance is the central contribution of the paper, but the theorem and the discussion around it is very imprecise which makes it difficult to use these results to develop a concrete understanding of the dynamics of SGD. It would be useful to dig deeper into these calculations to glean more actionable information from them.

**Justification For Why Not Higher Score:**

Please see above.

**Justification For Why Not Lower Score:**

N/A

---

### Decision · Program_Chairs · 2024-01-16

Reject